# Understanding Pruning at Initialization: An effective node-path balancing perspective

## Abstract

Pruning at initialization (PaI) methods aim to remove weights of neural networks before training in pursuit of reducing training costs. While current PaI methods are promising and outperform random pruning, much work remains to be done to understand and improve PaI methods to achieve the performance of pruning after training. In particular, recent studies (Frankle et al., 2021; Su et al., 2020) present empirical evidence for the potential of PaI, and show intriguing properties like layerwise random shuffling connections of pruned networks preserves or even improves the performance. Our paper gives new perspectives on PaI from the geometry of subnetwork configurations. We propose to use two quantities to probe the shape of subnetworks: the numbers of effective paths and effective nodes (or channels). Using these numbers, we provide a principled framework to better understand PaI methods. Our main findings are: (i) *the width of subnetworks matters* in regular sparsity levels ($< 99\%$) - this matches the competitive performance of shuffled layerwise subnetworks; (ii) *node-path balancing plays a critical role* in the quality of PaI subnetworks, especially in extreme sparsity regimes. These innovate an important direction to network pruning that takes into account the subnetwork topology itself. To illustrate the promise of this direction, we present a fairly naive method based on SynFlow (Tanaka et al., 2020) and conduct extensive experiments on different architectures and datasets to demonstrate its effectiveness.

## 1 Introduction

Deep neural networks have achieved state-of-the-art performance in a wide range of machine learning applications (Brown et al., 2020; Dosovitskiy et al., 2021; Ramesh et al., 2021; Radford et al., 2021). However, the huge computational resource requirements limit their applications, especially in edge computing and other future smart cyber-physical systems (Hinton et al., 2015; Zhao et al., 2019; Price & Tanner, 2021; Yuan et al., 2021; Bithika et al., 2022). To overcome this issue, there has been a number of attempts to reduce the size of such networks without compromising their performance, among which *pruning* enjoys a significant interest (Hoefler et al., 2021; Deng et al., 2020; Cheng et al., 2018). A rationale for this direction is the work of Frankle & Carbin (2018), in which the authors provide empirical evidences on the existence of sparse subnetworks that can be trained from scratch and achieve similar performance to the original network, referred to as the Lottery Tickets. However, standard methods for finding such subnetworks typically involve the costly pre-training and iterative magnitude pruning process.

This issue raises an intriguing research question: How to identify sparse, trainable subnetworks at initialization without pre-training? Specifically, a successful pruning before training method can significantly reduce both the cost of memory and runtime, without sacrificing performance much (Wang et al., 2022). This would make neural networks applicable even in scenarios with scarce computing resources (Alizadeh et al., 2022; Yuan et al., 2021). As such, many methods for PaI have been proposed (Lee et al., 2019; Tanaka et al., 2020; de Jorge et al., 2021; Wang et al., 2020; Alizadeh et al., 2022). While these methods are based on a number of intuitions (e.g., leveraging the gradient information), they typically measure the importance of network parameters. More recently, Frankle et al. (2021); Su et al. (2020) observe a rather surprising phenomenon: for PaI methods, layerwise shuffling connections of pruned subnetworks does not reduce the network's performance. A surprising consequence is that layerwise sparsity ratios are more important than weight-level importance scores of the subnetwork (Frankle et al., 2021). This indicates that in searching for good subnet-

works at initialization, the topology of subnetworks, in particular the number of input-output paths and active nodes, plays a vital role and should be investigated more extensively.

While the findings of previous works Frankle et al. (2021); Su et al. (2020) indicate that PaI methods are insensitive to random shuffling, we find this is not true in the extreme sparsity regime ($> 99\%$) in which the number of effective connections in subnetworks are vulnerable to changes to shuffling. In layerwise shuffling experiments (see Section 3.3), shuffling connections results in more effective nodes but substantially fewer input-output paths. In normal sparsity levels, shuffling weights in regular sparsities can maintain and even increase effective parameters (Frankle et al., 2021) and the number of activated nodes (Patil & Dovrolis, 2021). Having more effective nodes after shuffling helps the representation capacity of subnetworks enhance while the number of effective paths is enough for preserving the information flow, leading to competitive or even better performance of shuffled subnetworks (Section 3.3). However, we empirically show that in the extreme sparsity levels, while shuffling still more or less preserves the number of remaining attached nodes, the performance of shuffled subnetworks may drop significantly, compared to the unshuffled counterpart. This is because the information flow is hampered due to the significant decrease of the input-output paths. These findings suggest that separately considering effective paths or nodes is inadequate to fully capture behaviors of subnetworks generated by PaI methods.

In addition, we design a simple toy experiment in which an MLP network is considered as a base architecture. We then randomly generate 100 subnetworks at the same sparsity level $95\%$ while ensuring that all nodes in input and output layers are activated, all networks are trained to converge with the same setting and tested on MNIST dataset (more details in Appendix I). As depicted in Figure 1, these subnetworks have different effective nodes and paths. We observe that subnetworks with higher in both the number of nodes and paths tend to have better performance. This highlights the essential role of simultaneously considering both node and path in the success of designing subnetworks at initialization.

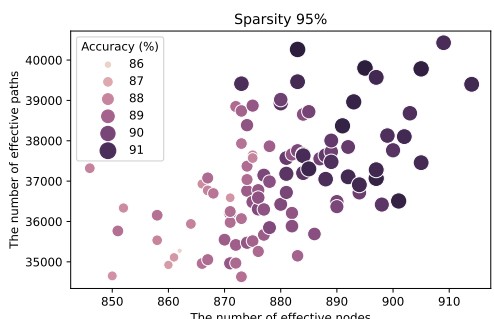

Figure 1: Toy experiment with 100 subnetworks randomly generated from the base MLP network.

To overcome this issue, we introduce a novel framework that combines metrics mentioned in previous works (Tanaka et al., 2020; Patil & Dovrolis, 2021; Frankle et al., 2021) to provide a more accurate explanation of the performance of different approaches. In particular, we propose the joint usage of both the number of input-output paths (a.k.a. effective paths) and activated (i.e., effective) nodes to explain different behaviors of PaI methods in a more comprehensive way (see Section 3.3 for more details). We also demonstrate the usefulness of this framework as follows: With a simple modification in the base of the iterative pruning algorithm, we show that if we maintain both the effective path and node level high simultaneously, the quality of subnetworks will be enhanced.

In summary, our main contributions are:

- We propose to systematically use the topology of subnetworks, particularly the number of effective nodes and paths, as proxies for the performance of PaI methods (Section 3). We revisit the layerwise shuffling sanity check on subnetworks produced by existing PaI methods and provide unified explanations for their behaviors based on these metrics in a wide range of sparsities (Section 3.3).

- We discover a new relation between the proposed metrics and the performance of subnetworks, termed the Node-Path Balancing Principle, that suggests a nontrivial balance between nodes and paths is necessary for optimal performance of PaI methods (Section 4). We introduce a simple modification of SynFlow (Tanaka et al., 2020) to give a proof-of-concept for our principle. We perform extensive experiments to show that better balancing this trade-off in pruned networks leads to improved performance (Section 5).

- Our framework opens a novel research direction that advocates taking into account both paths and nodes in the design of PaI methods. More precisely, while the regular sparsity regime places effective nodes in a higher priority than effective paths, extremely sparse subnetworks demand a more delicate balance between these two metrics (Section 5).

## 2 RELATED WORK

**Neural Network Pruning.** Neural network pruning methods (LeCun et al., 1989; Hassibi et al., 1993; Han et al., 2015) traditionally focus on pruning trained models based on pre-defined criteria, and then resulting subnetworks will be fine-tuned to converge. Recently, Frankle & Carbin (2018); Frankle et al. (2020) empirically show the existence of randomly initialized subnetworks (lottery tickets) which when trained from scratch or in early training iterations, that can achieve competitive performance with their dense counterparts. Unfortunately, finding lottery tickets is highly overhead due to the train and prune cycle. Gradual pruning methods (Zhu & Gupta, 2017; Gale et al., 2019) interleave the pruning and training, which are usually cheaper than pruning after training, but the network still needs to be trained to choose the ideal sparse subnetwork. Pruning before training methods (Lee et al., 2019; Wang et al., 2020; Patil & Dovrolis, 2021; Tanaka et al., 2020; Alizadeh et al., 2022) determine subnetworks by the network initialization, gradient information, and network topology. However, experimentally results done by Frankle et al. (2021); Su et al. (2020) show that proposed PaI criteria are not essential for obtaining a subnetwork with good performance.

**Pruning and network shape.** Since PaI methods do not utilize training data or use only negligible portions of data to obtain gradient information without training, the configuration of nodes and connections is an essential source of information for optimizing the performance of pruned networks. It turns out that some PaI methods implicitly optimize certain aspects of network shape. In particular, SynFlow(Tanaka et al., 2020) preserves the number of input-output paths as synaptic strength, but often creates more isolated neurons in pruned networks. The works of Patil & Dovrolis (2021) and Gebhart et al. (2021) aim to preserve proxies in terms of path kernels which are also directly related to the shape of subnetworks. Furthermore, while PHEW (Patil & Dovrolis, 2021) additionally implements random walks to increase the number of effective nodes, it unintentionally decreases the number of input-output paths. Our new point of view on node-path balancing would be helpful to systematically optimize network configuration for better performance. Other works also consider the number of effective nodes and effective paths to capture the capacity of pruned subnetworks (Wang et al., 2020; Naji et al., 2021) where these numbers are considered separately.

**Extreme Sparse Network.** In the context of extreme sparsity (Cho et al., 2021; Price & Tanner, 2021), the network density is less than $1\%$. Cho et al. (2021) associate two works from Lee et al. (2019); Zhou et al. (2019) for learning the masking during training. Tanaka et al. (2020); de Jorge et al. (2021); Vysogorets & Kempe (2021) leverage iterative pruning to keep subnetworks from layer collapse in super sparsity cases. Price & Tanner (2021) only require an extremely small amount of trainable parameters associated with a freeze fully connected network, which helps the model performs well on extreme sparsity settings. This suggests that preserving information flow through network connections plays a crucial role in intense sparse networks.

## 3 METHODOLOGY

### 3.1 PRUNING AT INITIALIZATION METHODS

Given a neural network, we first devide the network into layers: $\ell = 0$ is the input layer, then for each layer $\ell \in \{1, 2, \dots, L\}$, we flatten the weights on the connections from layer $\ell - 1$ to layer $\ell$ into a weight vector $w_\ell \in \mathbb{R}^{d_\ell}$, where $d_\ell$ is the number of connections from layer $\ell - 1$ to layer $\ell$. Let $\mathbf{w} = (w_1, \dots, w_L)$ denote the set of weights. Pruning generates binary mask vectors $m_\ell \in \{0, 1\}^{d_\ell}$ yielding sparse neural networks with sparse weights $m_\ell \odot w_\ell$ - the elementwise product of masks and weights. Sparsity is defined as the fraction of weights being removed: $s = 1 - \frac{\sum m_\ell}{\sum d_\ell} \in [0, 1]$.

A pruning method usually consists of two operations: *score* and *remove*, where *score* takes as input weights of the network, and outputs an important score for each weights: $z_\ell = score(w_\ell) \in \mathbb{R}^\ell$; then *remove* takes as input the scores $\mathbf{z} = (z_1, \dots, z_L)$ and the sparsity $s$, and outputs the masks $m_\ell$ with overall sparsity $s$. Pruning can be done in one-shot or iteratively. For one-shot pruning, we only generate the scores once, then prune the network upto sparsity $s$. For iterative pruning, we repeat the processes of score, then prune from sparsity $s^{(t-1)/T}$ to sparsity $s^{t/T}$ iteratively $T$ times.

**Random.** This method assigns each connection with a random score from a uniform distribution $\mathcal{U}(0, 1)$. Random pruning empirically prunes each layer to target sparsity $s$ (Liu et al., 2022).

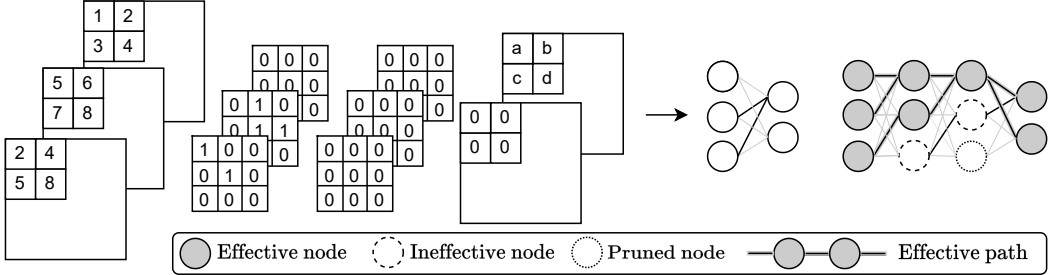

Figure 2: An example of effective paths and effective nodes.

**SNIP.** SNIP was introduced by Lee et al. (2019) with the pruning objective of reducing connection sensitivity to the training loss. One passes a mini-batch of data through the network and compute the score $\mathbf{z}$ for weight $\mathbf{w}$ of SNIP as $\mathbf{z} = |\mathbf{w} \odot \nabla_{\mathbf{w}} \mathcal{L}|$.

**Iterative SNIP.** This is an iterative variant of SNIP (de Jorge et al., 2021) with the same important score. But, iterative SNIP gradually prunes the remaining weights with lowest scores from sparsity $s^{\frac{t-1}{T}}$ to sparsity $s^{\frac{t}{T}}$ iteratively $T$ times for $t = 1, 2, \ldots, T$.

**SynFlow.** SynFlow (Tanaka et al., 2020) is an iterative and data-agnostic PaI method. The pruning objective of SynFlow is to make the network remains connected until the most extreme possible sparsity. The weight scores are computed as follows. One first replaces all weights in the network by their absolute values. Then, an $\mathbf{1}$ input tensor is passed through the network, and one computes the sum of the logits as $R = \mathbf{1}^{\top}(\prod_{\ell=1}^{L} |w_\ell|)\mathbf{1}$. Finally, the score of weight $\mathbf{w}$ is computed as $\mathbf{z} = |\mathbf{w} \odot \nabla_{\mathbf{w}} R|$. SynFlow prunes the network iteratively $T$ times.

**PHEW.** PHEW (Patil & Dovrolis, 2021) is also an iterative and data-independent PaI method. It selects a set of input-output paths to be preserved. These paths are chosen through random walks, biased towards higher-weight magnitudes. The selection start from a unit that is selected through round robin procedure. This process continues until the subnetwork achieves the predefined sparsity.

### 3.2 METRIC DEFINITION

In a sparse network, it is intuitively clear that one should arrange the connections into a configuration neither too thin nor too spread-out to have good information propagation during training. For a better measurement, we propose using two metrics to evaluate the quality of subnetworks. Please refer to Appendix B for detailed discussions and Python code for calculating the metrics.

**Effective path.** We define a path to be effective if it connects an input node to an output node without interruption (see Figure 2). Metrics based on paths are mentioned in Tanaka et al. (2020); Gebhart et al. (2021) as $l_1$ and $l_2$ path norms, respectively. In this paper, we only take into account the number of paths.

**Effective node/channel.** An effective node/channel is one that at least one effective path goes through it (demonstrated as the right part in Figure 2). This concept is also considered in works of Patil & Dovrolis (2021); Frankle et al. (2021). For convolutional layers, we consider a kernel as a connection, and a channel as a node, and then convert the convolutional layer into a fully connected layer. An illustrative example can be found in Figure 2.

In the following sections, we compute and visualize the ratio of these metrics. In particular, we take logarithm scale of the number of effective paths and calculate the ratio of the pruned network and the dense ones, since effective paths have very large values. With effective nodes, we simply compute the ratio of the number of effective nodes between after and before pruning.

### 3.3 LAYERWISE SHUFFLING PHENOMENON

In this section, we investigate the intriguing phenomenon that layer-wise reshuffling the subnetwork found by PaI methods still produces competitive accuracy (Frankle et al., 2021; Su et al., 2020). Based on metrics, we provide a new way to understand why reshuffling subnetworks work and when they fail. We first use three PaI methods, i.e., SNIP (Lee et al., 2019), SynFlow (Tanaka et al., 2020), and PHEW (Patil & Dovrolis, 2021), to find the subnetworks. Then, we randomly shuffle

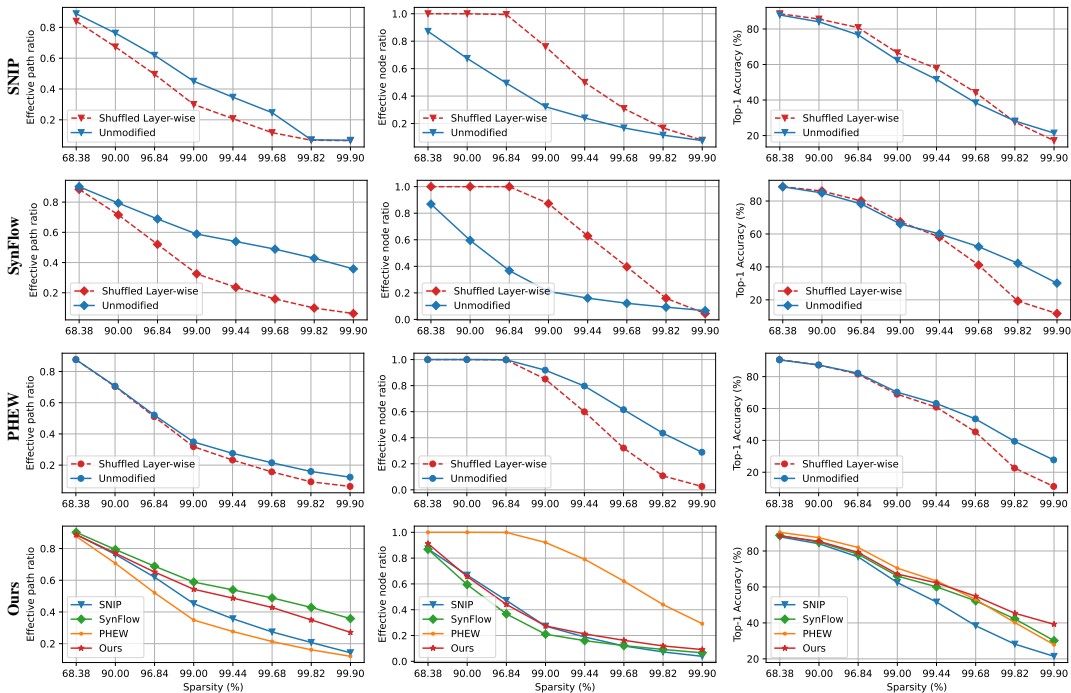

Figure 3: Layerwise shuffling results on various sparse subnetworks of ResNet20 produced by SNIP, SynFlow, and PHEW at initialization on CIFAR-10.

the pruning mask $m_l$. All the subnetworks (both unmodified and shuffled) are trained in the same setting. Finally, we calculate metrics and visualize the average scores in Figure 3.

In SNIP and SynFlow, the number of effective nodes drops significantly as sparsity increases, especially SynFlow by using iterative pruning procedure. After reshuffling, the connections are distributed uniformly in each layer. As a result, the number of effective nodes increases leading to essentially wider subnetworks while the number of effective paths decreases. In contrast, PHEW focuses on increasing the number of effective nodes by gradually adding new paths such that the network is as wide as possible. Consequently, reshuffling hurts the concrete network configuration, then reduces both the number of effective nodes and effective paths as sparsity is higher.

In normal sparsity cases, layerwise shuffled subnetworks show competitive performance or even better (with SynFlow and SNIP) compared with unmodified ones. In addition to the explanation based on the effective connection preservation from Frankle et al. (2021), we provide a more comprehensive heuristic reason to understand this phenomenon. In particular, at these sparsities (below ratio 99%), shuffled subnetworks with SynFlow and SNIP still ensure the information flow from input to output through enough effective paths, meanwhile the representation capacity of subnetworks is increased due to the growth in the number of effective nodes. In more detail, Figure 3 shows that the number of effective nodes and effective paths of shuffled layerwise subnetworks produced by SynFlow and SNIP are similar to unmodified subnetworks of PHEW with corresponding sparsities. It suggests that the effective network width is more vital in normal sparsity when the number of input-output paths is still enough to transfer information.

However, in the extreme sparsity regime, random shuffling significantly reduces the performance of the subnetworks. Along with the reduction of the number of effective parameters (see Appendix C), the number of input-output paths decreases substantially when the network becomes more sparse. Even though layerwisely shuffled networks become wider, the limited number of effective paths hinders the information flows in the subnetworks. These explain why the accuracy of shuffled subnetworks is much lower than the unmodified ones in intensive sparsities.

These observations indicate that increasing the number of effective paths (SynFlow) or effective nodes (PHEW) alone is not sufficient in the design of PaI methods. We hypothesize that to have better subnetworks, the number of effective paths and effective nodes should be concurrently con-

sidered. If we balance these two metrics well, the performance after training of subnetworks will be enhanced, especially in extreme sparsity levels (see the last row of Figure 3).

## 4    NODE-PATH BALANCING PRINCIPLE

From observations in Section 3.3 and toy example in Appendix I, both effective paths and nodes have shown their critical roles in the performance of subnetworks. We now formally state the *Node-Path Balancing Principle*.

**Node-Path Balancing Principle:** The combination of both the numbers of effective nodes and effective paths is a proxy to potential performance of subnetworks under pruning. A pruning at initialization method which produces pruned subnetworks with too many effective paths (resp. effective nodes) will has less than necessary the number of effective nodes (resp. effective paths), and consequentially has suboptimal performance. It is necessary to balance the numbers of effective nodes and effective paths for better performance of pruned subnetworks.

---

**Algorithm 1** Iterative pruning algorithm to find a pruning mask

---
1:  **Inputs:** Final sparsity $s$, number of pruning steps $T$, weights $\mathbf{w}_{init}$, $\Delta_t$, $T_{max}$
2:  Obtain sparsity $\{s_t\}_{t=1:T}$ at each pruning iteration
3:  Define intial mask $\mathbf{m}_0 = \mathbf{1}$
4:  $\mathbf{w}_0 = \mathbf{w}_{init} \odot \mathbf{m}_0$
5:  **for** $t = 0, \dots, T-1$ **do**
6:      **if** $t \% \Delta_t = 0$ and $T_{max} < t$ **then**
7:          Find $\mathbf{m}_t$ by Random pruning with sparsity $s_t$
8:      **else**
9:          Find $\mathbf{m}_t$ by SynFlow pruning with sparsity $s_t$
10:     **end if**
11:     $\mathbf{w}_t = \mathbf{w}_{t-1} \odot \mathbf{m}_t$
12: **end for**
13: **Return:** $\mathbf{w}_T$

---

**Proposed Method.**   Here, we describe a modification of the iterative pruning algorithm that better satisfies the Node-Path Balancing Principle. We choose SynFlow (Tanaka et al., 2020) as the base method since it solely focuses on optimizing effective paths, hence, the node-path balancing effect, resulted from our modifications, will be easy to observe. Leveraging the iterative pruning procedure of SynFlow, we introduce a new pruning scheduler to balance effective nodes and paths by performing random pruning at several pruning iterations. We arbitrarily replace the pruning criterion in some early iterations with random pruning. The justification for doing this is based on the intuition that in iterative pruning methods like SynFlow if a node has some pruned connections (low-degree node) in the initial few iterations it has a higher likelihood of being detached in later iterations (Patil & Dovrolis, 2021). Therefore, our method helps low-degree nodes avoid being removed at subsequent iterations since it randomly eliminates connections to high-degree units.

Specifically, we introduce two additional hyperparameters $\Delta_t$ and $T_{max}$, which means in the first $T_{max}$ pruning iteration, we use random pruning after each $(\Delta_t - 1)$ steps. The pseudo-code is in Algorithm 1. Adjusting these two hyperparameters leads to different configurations of subnetworks. Intuitively, a large $T_{max}$ easily drives subnetworks to layer-collapse since the pruner randomly removes important connections when the network is in intensive sparsity, which destroys a large number of input-output paths. While randomly pruning the network when it is in normal sparsities may increase the subnetwork's width because it fortuitously prunes edges to high-degree nodes. More investigations on pruning scheduler are provided in Section 5.4 and Appendix G.

## 5    EXPERIMENT

### 5.1    EXPERIMENTAL SETTINGS

**Architectures and Datasets.**   Our main experiments are conducted with CIFAR-10 and Tiny-Imagenet datasets. With CIFAR-10, we use ResNet-20 and VGG-19 from  Tanaka et al. (2020). For Tiny-ImageNet, we utilize ResNet-18 with 18 layers adapted from Tanaka et al. (2020). We

treat all of the weights from convolutional and linear layers of these networks are prunable parameters, but we do not prune the biases nor the weights in the batch normalization layers. We run five seeds with CIFAR-10 experiments and three seeds with experiments on Tiny-Imagenet. We take the average value of the results and plot them in visualizations. More details are in Appendix A.

## 5.2 EXPERIMENT ON REGULAR SPARSITIES

In Figure 4, we show the results of different PaI methods with varied conventional sparsities on three settings: VGG-19 on CIFAR-10, ResNet-20 on CIFAR-10 and ResNet-18 on Tiny-Imagenet. In all settings, we use $\Delta_t = 2$ and $T_{max} = 15$ for our method. Our main observations are below.

*Effective node is more important in the normal sparsity regime.* In normal sparsities ($< 99\%$), PHEW, a data-agnostic method, consistently performs better than other data-independent methods like SynFlow and Random, as well as data-dependent methods like SNIP and Iter-SNIP. The second column in Figure 4 shows that PHEW produces wider subnetworks associated with adequate number of input-output paths (ratios $> 40\%$) which attributes to its superior performance. Specifically, broader width enhances the representation potential of the subnetworks while having enough effective paths preserves the information flow in the network Patil & Dovrolis (2021). These two factors together contribute to efficient training of pruned subnetworks leading to higher accuracy.

*With the same number of effective nodes, the higher the number of effective paths, the better performance subnetworks will reach after training.* In Tiny-Imagenet experiments, our methods, Random, and PHEW produce subnetworks with all nodes activated up to sparsity $99.00\%$. However, subnetworks generated by our method have slightly higher effective paths than PHEW and are significantly higher than that of Random, which results in improved performances. This behavior also happens in experiments with ResNet-20 where our method and SNIP have the same number of activated units but Ours has more input-output paths.

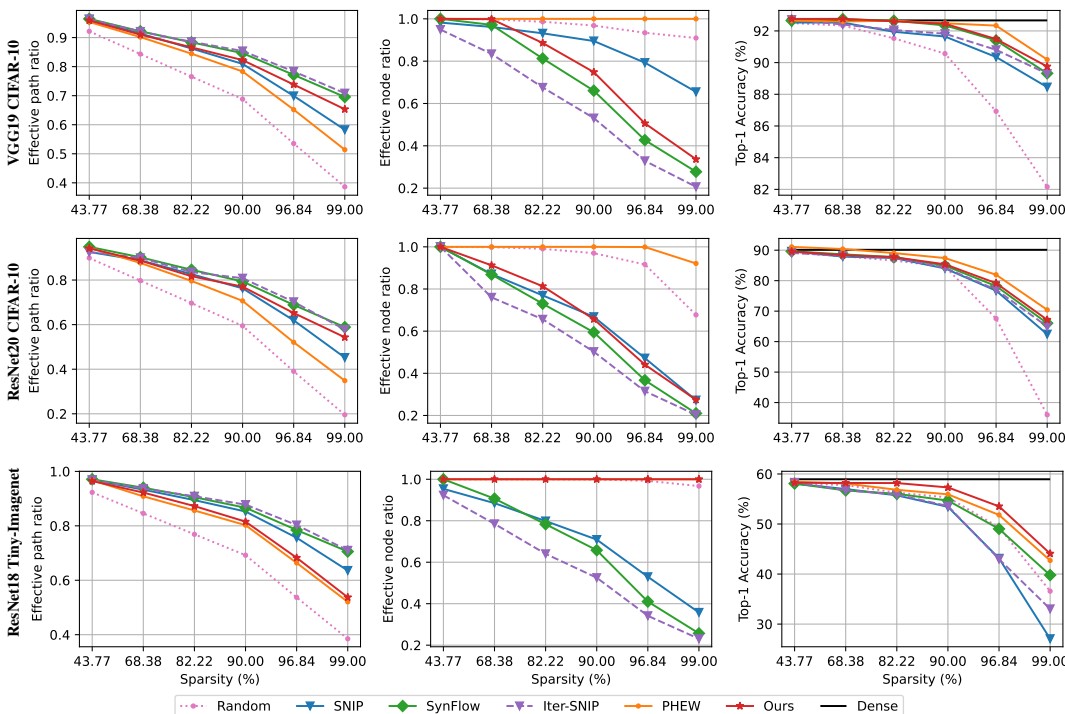

Figure 4: The ratio of effective paths (after log scale), nodes after and before pruning, and the corresponding accuracy of different PaI methods on three datasets in the regular sparsity regimes.

## 5.3 EXPERIMENT ON EXTREME SPARSITIES

In Figure 5, we show the results of different PaI methods with intensive sparsities. We summarize our main observations below.

*Extremely sparse subnetworks prefer effective paths to effective nodes.* In extreme sparsities ($> 99\%$), with very limited parameters, despite the higher effective nodes, PHEW, SNIP, and Random prunings start performing poorly than iterative pruning methods like SynFlow and Iter-SNIP (see the first two rows in Figure 5). We hypothesize that it is because both PHEW and Random pruning produce small numbers of input-output paths (the effective path ratios are below $10\%$) which makes the pruned subnetworks contain great number of straight paths (lacks criss-cross connections). This hinders the optimization of subnetworks. However, if we better balance the trade-off between effective nodes and effective paths, the accuracy increases. For example, with experiments on CIFAR-10, our method increases subnetworks width compared with SynFlow and Iter-SNIP while the number of effective paths is much higher than PHEW. Therefore, the performance of pruned networks after training is enhanced. Further discussions can be found in Appendix D.

*With similar numbers of effective paths in extreme sparsities, subnetworks that have more effective nodes show better performance.* We want to highlight that it is intractable to force two methods to generate subnetworks with the same number of effective paths given a sparse ratio. Therefore, we select some subnetworks having similar input-output paths in experiments. In experiments with ResNet-20 on CIFAR-10 (the second row in Figure 5), our method and Iter-SNIP produce two subnetworks with similar numbers of effective paths in sparsities from $99.00\%$ to $99.90\%$. However, our method produces broader subnetworks, which induces better performance.

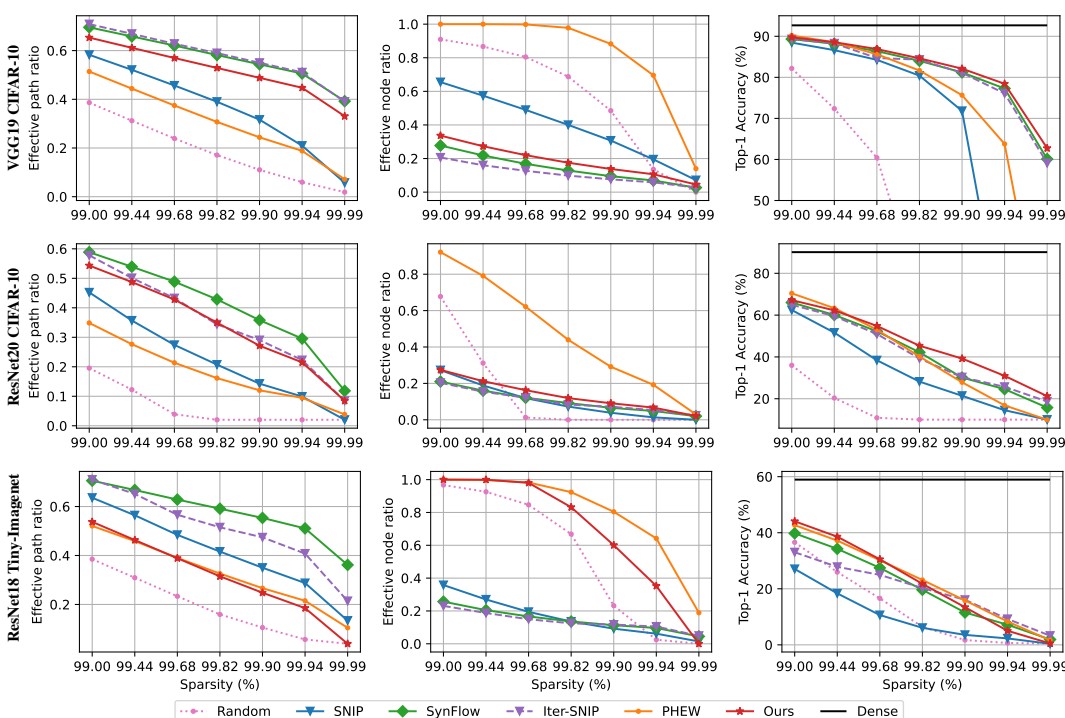

Figure 5: The ratio of effective paths (after log scale), nodes after and before pruning, and the corresponding accuracy of different PaI methods on three datasets in the extreme sparsity regimes.

The above experimental results bolster our Node-Path Balancing Principle. In particular, these findings indicate that pruning neural networks in the regular sparsity regime should give more consideration to the number of effective neurons since the information flow is conserved by the sufficient number of input-output paths. Meanwhile, when the sparsity level becomes more extreme, a delicate balance between the number of paths and nodes leads to gains in performance of subnetworks.

## 5.4 EXPERIMENTAL RESULTS WITH DIFFERENT PRUNING SCHEDULERS

In this part, we investigate how different choices of the scheduler in our method affect the performance of subnetworks through two hyperparameters: $\Delta_t$ and $T_{max}$. We first vary $T_{max} = \{10, 15\}$, $\Delta_t = \{2, 3\}$ to add more random pruning in early iterations with the goal of avoiding layer-collapse in extremely sparse ratios. We also change the value of $T_{max} = \{15, 20, 30, 40, 50, 60, 70, 80, 90\}$

with the fixed $\Delta_t = 5$ to investigate how $T_{max}$ affects the subnetwork's structure. We visualized selected results in Figure 6 and more in Appendix G. Our main observations are as follows.

*Applying random pruning during the pruning process creates large-width subnetworks in regular sparsity levels.* In SynFlow, if a neuron loses some connections in previous pruning iterations, it has a high probability to be pruned in subsequent iterations as result of the decrease in the number of pathways that pass through that unit. Therefore, higher $T_{max}$ allows the pruner to remove edges connecting to units that have more connections. This implicitly constructs wider subnetworks as shown in Figure 6 and Appendix G.

*In extreme sparsity regimes, a large $T_{max}$ significantly reduces the number of effective paths.* When the subnetwork achieves a specific high sparsity, randomly removing connections in the subnetwork is more likely to destroy effective paths. This is presented in Figure 6 when $T_{max} = 50$ or even more severe with higher $T_{max} = 80$.

*Frequently using random pruning in the early stage of pruning generates subnetworks with better performance in extreme sparsities.* At the first few pruning iterations, subnetworks have high density, thus, randomly eliminating edges has less influence on preventing substantial drop of the number of effective paths. Moreover, it has effects on growing the chance of being kept for low-degree nodes. This is supported by the experimental results of red and blue lines in Figure 6.

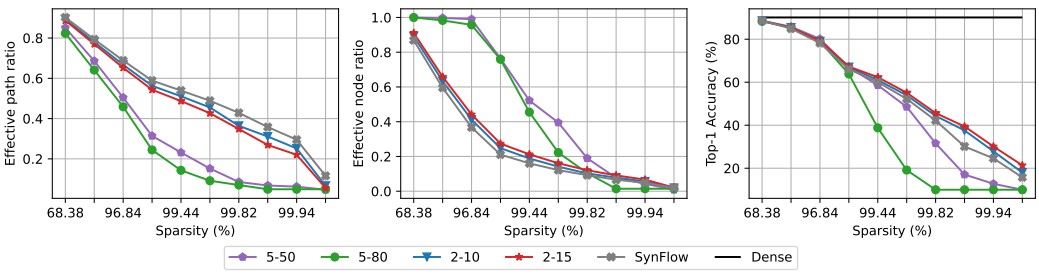

Figure 6: Scheduler ablation with ResNet-20 on CIFAR-10. We vary the value of $\Delta_t$ and $T_{max}$ and name schedulers as $\Delta_t$-$T_{max}$.

## 6    CONCLUSION

In this work, we propose a new framework to study PaI methods via systematic uses of the configuration of pruned subnetworks based on two different metrics: the number of effective paths and the number of effective nodes of subnetworks. As proxies to performance of pruned networks of PaI methods, we use these two metrics in combination. We discover a new relation between these metrics, called Node-Path Balancing Principle, which guides the understanding and optimization of PaI methods. We then propose a simple method to demonstrate that nontrivial balancing of numbers of effective paths and nodes can lead to improved performance of PaI methods. Our framework provides unified explanations for the intriguing Layerwise Connection Reshuffling Phenomenon (Su et al., 2020; Frankle et al., 2021) of subnetworks produced by PaI methods in normal pruning sparsity regime, as well as the failure of this phenomenon in extreme sparsity levels. Our findings are supported by extensive experiments on different model architectures and datasets. Our new perspective based on configuration of subnetworks, in terms of effective nodes and effective paths, provide new insights to the working mechanism of PaI methods, and opens new research directions on neural network pruning methods, as well as designs of sparse neural network.

In particular, our framework directly calls for attention to optimizing the number of effective nodes and effective paths. Guiding by the Node-Path Balancing Principle, our goal is to optimize the number of activated nodes and paths at given sparsity levels without using data. A potential approach is to consider the pruning problem as a multi-objective optimization problem. More precisely, our problem becomes *given an architecture, find the optimal balance between effective paths and effective nodes with a given number of parameters*. This is a non-trivial problem, but with the advances from multi-objective optimization literature, there are many promising approaches.

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

## A  EXPERIMENT DETAILS

In this section, along with pruning at initialization (PaI) methods in the main text, we provide experimental results with GraSP (Wang et al., 2020). In particular, GraSP is another gradient-based pruning PaI that aims to preserve the gradient flow of sparse networks obtained by pruning. The score $z$ of weight $w$ in GraSP is computed as $\mathbf{z} = -\mathbf{w} \odot (\mathbf{H}\nabla_{\mathbf{w}}\mathcal{L})$, where $\mathbf{H}$ is the Hessian of the training loss after passing a mini-batch of data through the network.

We describe our experiment settings on architectures and datasets. We conduct experiments on a single GTX 3090Ti and use Pytorch library. We adapt Tanaka et al. (2020) source code [1] for SNIP, GraSP, SynFlow, and Random, and change the prune epochs to 100 for Iterative SNIP instead of 1 in SNIP. And we use the official code [2] of Patil & Dovrolis (2021) for PHEW.

**Datasets.** Our main experiments are conducted with CIFAR-10 and Tiny-Imagenet datasets, where:

- CIFAR-10 is augmented by normalizing per-channel, randomly flipping horizontally.
- Tiny-ImageNet is augmented by normalizing per channel, cropping to 64x64, and randomly flipping horizontally.

**Architectures.** We use three different networks:

- VGG-19 is a CIFAR-10 network used in SynFlow (Tanaka et al., 2020). We choose a batch-normalization version.
- ResNet-20 is a 20-layer CIFAR-10 version of ResNet created by He et al. (2016). This version has added batch normalization layers before each activation function.
- ResNet-18 is a ImageNet version with 18 layers adapted from SynFlow (Tanaka et al., 2020). The first convolution has kernel size 3x3 (instead of 7x7) and max-pooling layer that follows has been removed.

We treat all of the weights from convolutional and linear layers of these networks are prunable parameters, but we do not prune the biases nor the weights in the batch normalization layers. The weights in convolutional and linear layers are initialized with Kaiming normal, while biases are initialized to be zero. We run five seeds with CIFAR-10 experiments and three seeds with experiments on Tiny-Imagenet.

**Training details** With Iterative pruning methods SNIP and SynFlow, we use 100 pruning epochs. With methods using training data like SNIP, GraSP, and Iterative-SNIP, we randomly select 10 samples for each class, particularly, 100 data points for CIFAR-10, 2000 data samples for Tiny-ImageNet. Other hyperparameters are chosen as follow:

Table 1: Summary of the architectures, datasets, and hyperparameters used in experiments.

| Network | Dataset | Epochs | Batch | Optimizer | Momentum | LR | LR Drop, Epoch | Weight Decay |
|---------|---------|--------|-------|-----------|----------|-----|----------------|--------------|
| VGG-19 | CIFAR-10 | 160 | 128 | SGD | 0.9 | 0.1 | 10x, [60,120] | 0.0001 |
| ResNet-20 | CIFAR-10 | 160 | 128 | SGD | 0.9 | 0.1 | 10x, [60,120] | 0.0001 |
| ResNet-18 | Tiny-ImageNet | 100 | 128 | SGD | 0.9 | 0.01 | 10x, [30,60,80] | 0.0001 |

---

[1] https://github.com/ganguli-lab/Synaptic-Flow
[2] https://github.com/ShreyasMalakarjunPatil/PHEW

## B EFFECTIVE METRICS CALCULATION

**Effective path.** To exactly compute the number of effective paths, we remove the batch normalization layers, we initialize all the remaining parameters to 1. Then, we put the input vector one to the network, and the number of effective paths is the sum of logits on the output layer $R = \mathbf{1}^\top (\prod_{\ell=1}^{L} |w_\ell|) \mathbf{1}$.

More precisely, we face problems with pooling layers in convolutional neural networks. With max pooling layer, we simply do not modify the output of this layer. At that time, the result is the maximum number of paths in subnetworks. With average pooling layer, since all inputs of this layer contribute to the output, we change the average operator to the sum operator to exactly compute the number of effective paths. We all use ReLU activation functions in computing this metric since this function does not affect the results of calculations.

**Effective parameter.** We follow Frankle et al. (2021) when identifying which is effective parameter. Similar to computing effective paths, we make further steps. After having the sum of logits, we compute the gradients of this sum with respect to weights $\nabla_{\mathbf{w}} R$. Then, if an unpruned weight has a non-zero gradient, it is effective and vice versa. Effective parameters are dense edges that connect two effective nodes as visualized in Figure 2.

**Effective node/channel.** With fully connected layers, if all connections to one node or out of one node are pruned, this node is pruned node. If there exist connections to a node but all of these connections are ineffective, then this node becomes ineffective In convolutional layers, instead of nodes, we have channels. We consider a kernel as a connection, a channel as a node, and then convert the convolutional layer into a fully connected layer. The connection is pruned if and only if all parameters in the corresponding kernel are removed. Finally, identifying the effective nodes/channels is similar to the way in fully connected layers.

```python
def metric_calculation(model, mask):
    """
    model:  network architecture
    mask:   mask for subnetwork
    """
    n_eff_paths = 0
    n_eff_nodes = 0
    n_eff_params = 0

    # Initialize network with pruned weight = 0 and kept weight = 1
    for name, param in model.named_parameters():
        param.copy_(mask[name])

    x = torch.ones((1,c,h,w)) # c: channel - h: height - w: width
    y = model(x)
    sum_logits = y.sum()

    n_eff_paths = sum_logits.item()

    sum_logits.backward()
    with torch.no_grad():
        for name, param in model.named_parameters():
            eff_param = torch.where(param.grad.data!=0, 1, 0)
            n_eff_params += torch.sum(eff_param)

            eff_in_node = torch.where(torch.sum(eff_param,d=0)>0, 1, 0)
            n_eff_nodes += torch.sum(eff_in_node)

        # with output layer
        eff_out_node = torch.where(y>0, 1, 0)
        n_eff_nodes += torch.sum(eff_out_node)

    return n_eff_paths, n_eff_nodes, n_eff_params
```

Listing 1: Metric calculation example in fully connected neural networks

# C    LAYERWISE SHUFFLING EXPERIMENTS

With each setting, at each sparsity ratio, we seek subnetworks with 5 different seeds, and with each seed, we randomly shuffle the subnetwork two times. In addition to effective path ratios and effective node ratios, we compute the number of effective parameters after pruning and the actual remaining ones, then calculate the ratio between these two values.

Similar to Frankle et al. (2021) results, the performance and the number of effective parameters of high-density subnetworks after permuting the connections are similar to or even higher (in SNIP) than the unmodified ones. However, when the sparsity level becomes more intensive, the configuration of subnetworks is more concrete. Randomly rearranging connections within layers destroys this strict structure by detaching important edges, which drastically reduces the number of effective paths. The shuffled subnetworks lack input-output paths to transfer information during training, leading to a drop in performance compared with unmodified ones.

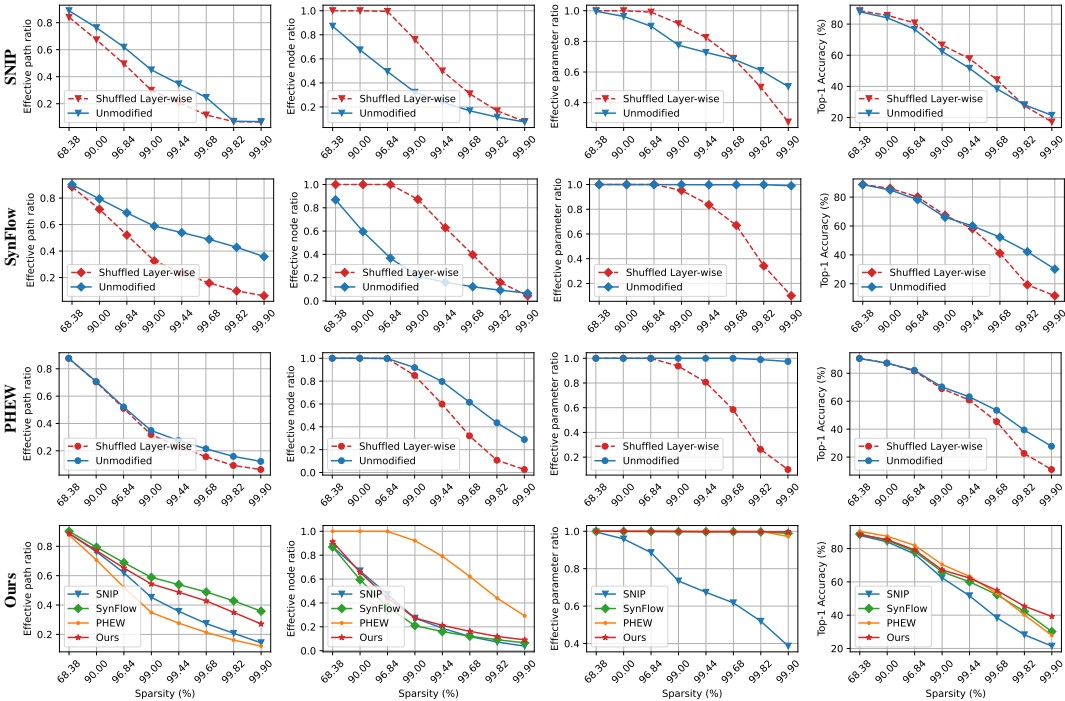

Figure 7: Layerwise shuffling results on various sparse subnetworks of ResNet20 produced SNIP, SynFlow, and PHEW at initialization on CIFAR-10.

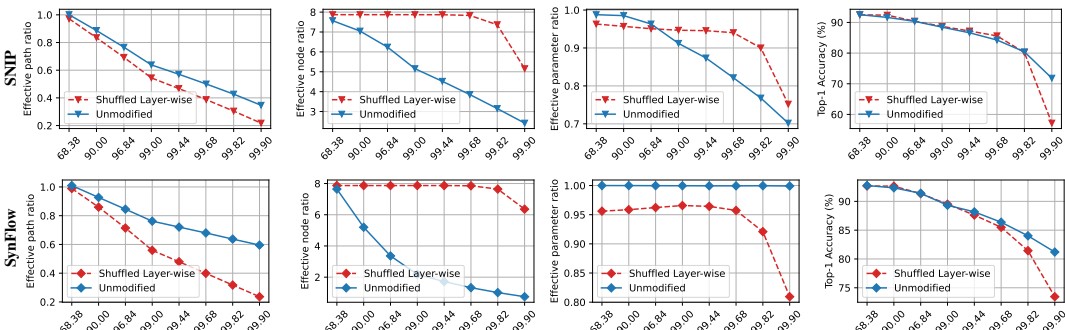

Figure 8: Layerwise shuffling results on various sparse subnetworks of VGG19 produced by SNIP an SynFlow at initialization on CIFAR-10.

# D    DIFFERENT SPARSITY REGIMES EXPERIMENTS INCLUDING GRASP

We report the min/average/max value of multiple runs and also the results of GraSP method in this Appendix to avoid ambiguity in the main text. Interestingly, GraSP tends to produce broader subnetworks compared with SNIP or SynFlow since GraSP's goal is preserving gradient flow after pruning (Wang et al., 2020). Iterative pruning methods like ours, SynFlow, and PHEW generate subnetworks with low variance in terms of number of effective paths and nodes and accuracy after training as well. When sparsity becomes more extremely, effective parameters (Figure 11) and effective paths (Figure 10) of one-shot pruning methods like SNIP, GraSP or Random drop significantly, which drives to decrease in the performance.

With ResNet-18 on Tiny-Imagenet experiments, PHEW and our method still preserve a relatively large number of effective paths ($> 30\%$) in extremely sparse ratios (up to sparsity $99.82\%$), which ensures the information transfers well in the sparse networks. Therefore, both perform better than path preserving methods like SynFlow and Iter-SNIP.

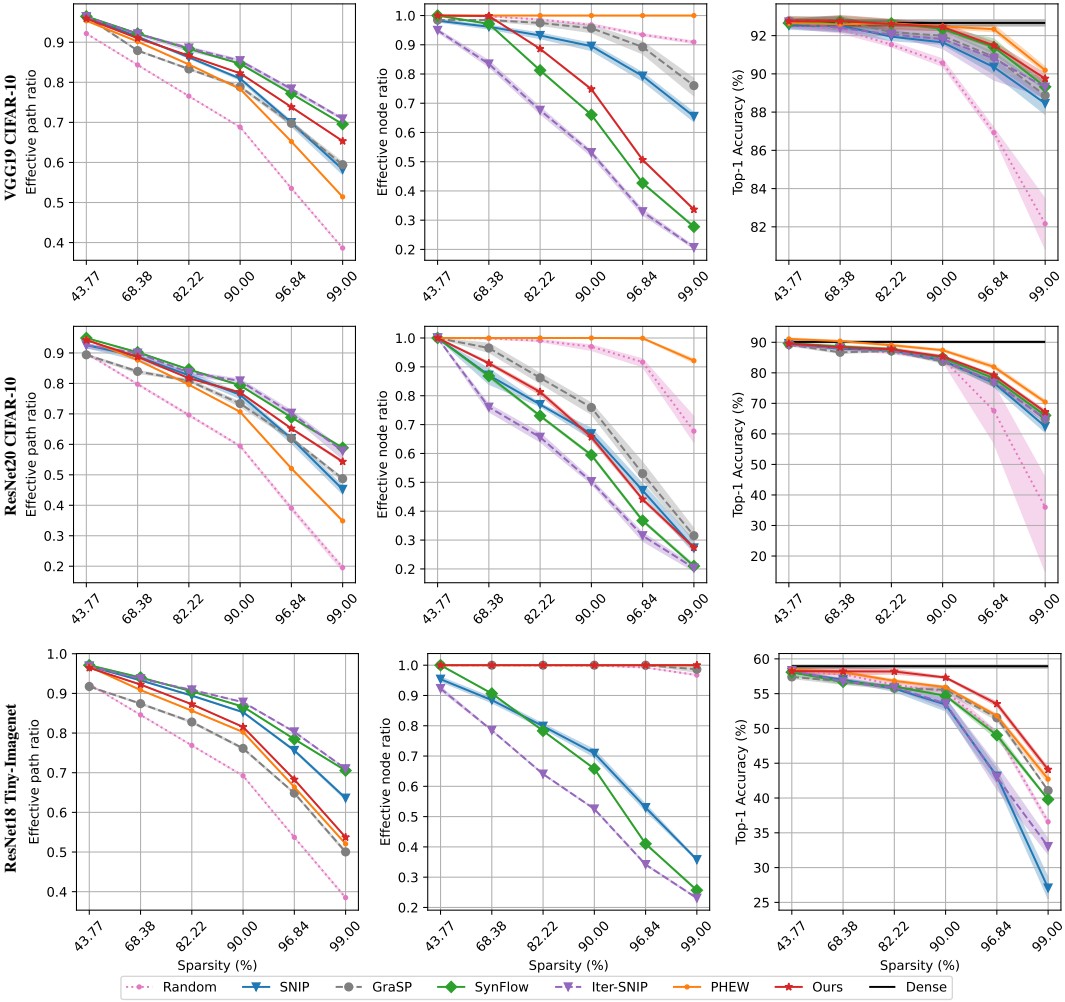

Figure 9: The ratio of effective paths (after log scale), nodes after and before pruning, and the corresponding accuracy of different PaI methods on three datasets in the regular sparsity regime.

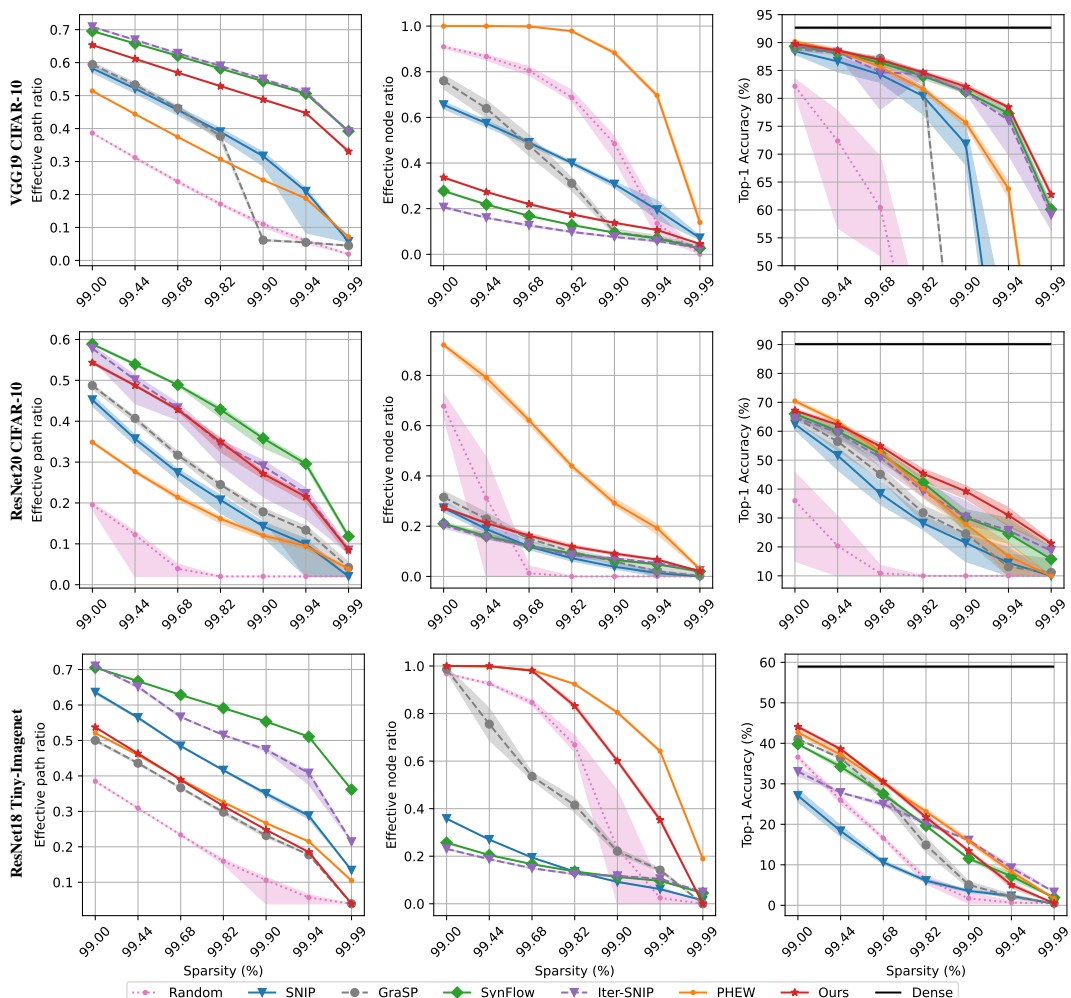

Figure 10: The ratio of effective paths (after log scale), nodes after and before pruning, and the corresponding accuracy of different PaI methods on three datasets in the extreme sparsity regime.

# E  EFFECTIVE PARAMETER RATIOS

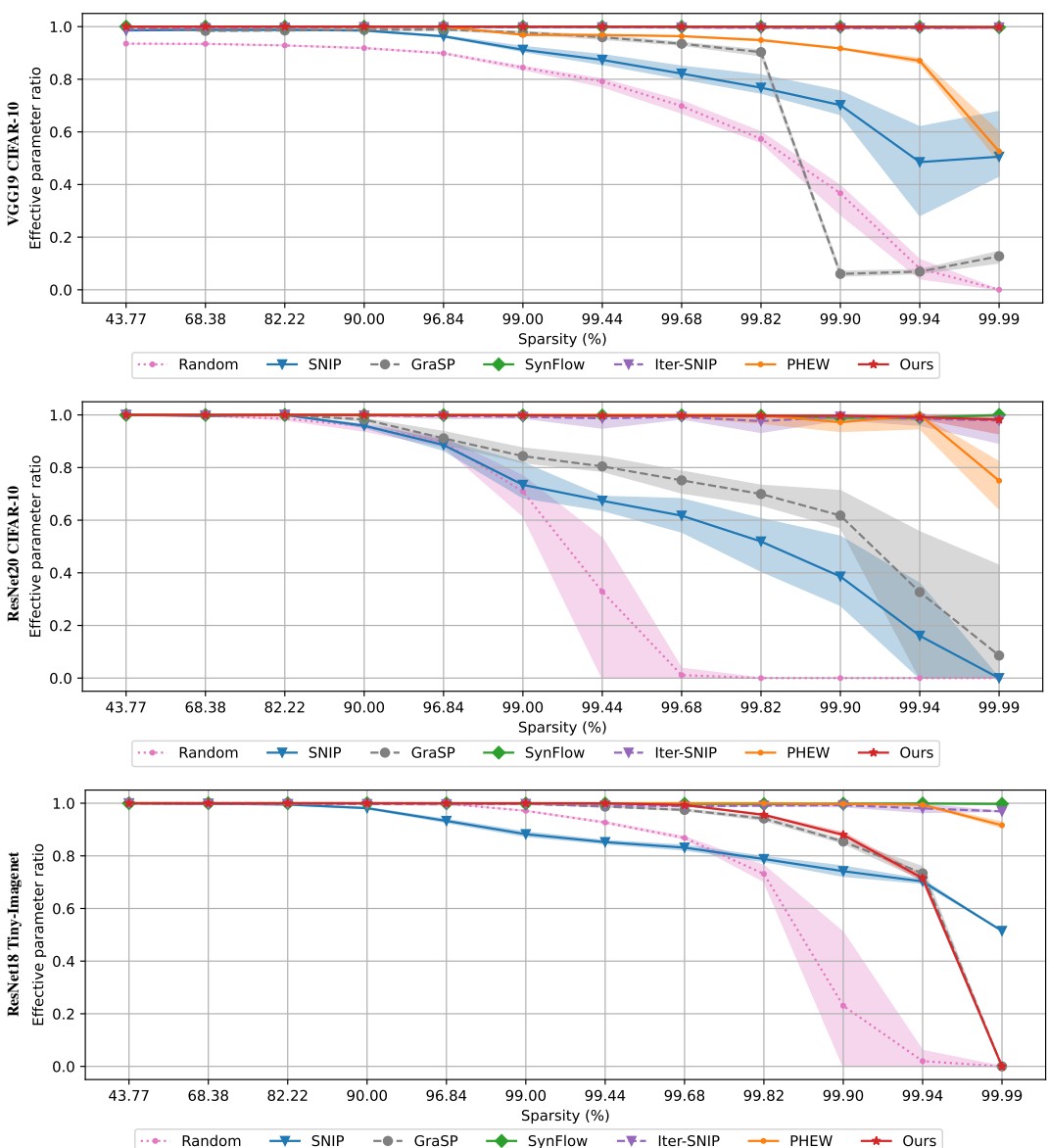

Figure 11: The ratio of effective parameters and unpruned parameters with different PaI methods on three datasets in varied sparsity regimes.

# F  EFFECTIVE NODES AT EACH LAYER

In Figure 12, we visualize the number of activated channels/nodes in each hidden layer of networks in different sparsities and settings. As mentioned, PHEW gradually adds new input-output paths such that nodes are activated as highly as possible. This is why PHEW consistently creates wider subnetworks. Except PHEW, iterative pruning methods (Iterative SNIP and SynFlow) tend to prune nodes more in later hidden layers compared with SNIP and GraSP because convolution layers at the top usually learn highly sparse features and thus more weights can be pruned (Wang et al., 2020; Zhang et al., 2019). We believe that further investigations into the configuration of hidden layers bring insightful knowledge to understand the success of PaI methods.

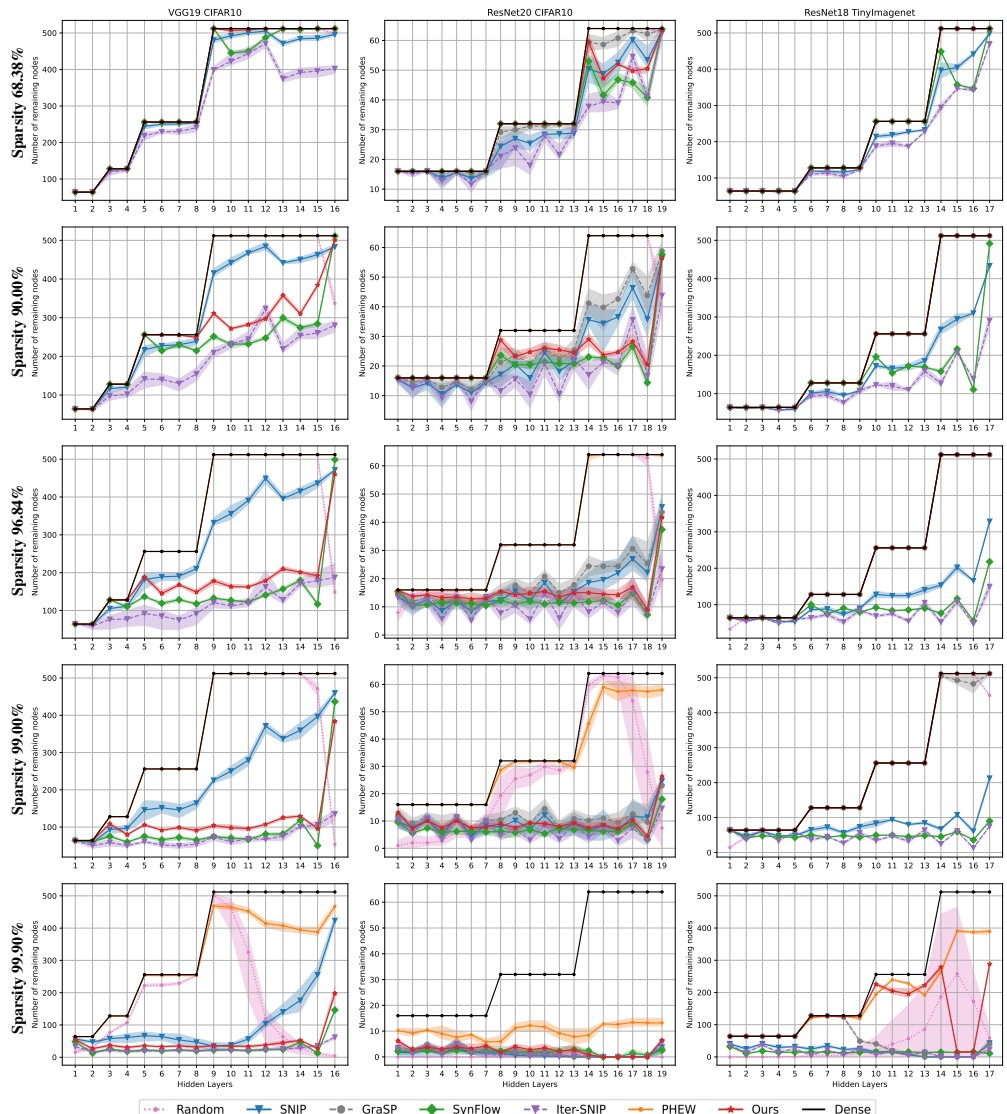

Figure 12: The number of effective nodes in each hidden layer of three settings on different sparsities.

# G ADDITIONAL RESULTS ON SCHEDULER

We present ablations on the scheduler with different settings. The observations are consistent in the main text. Higher $T_{max}$ tends to produce wider subnetworks in normal sparsities, but it drives to layer-collapse in extensive sparsities. These experimental results one more time highlight our Node-Path Balancing Principle that regular sparse subnetworks prefer effective nodes to effective paths while subnetworks with extreme sparsity levels require a good balance between nodes and paths.

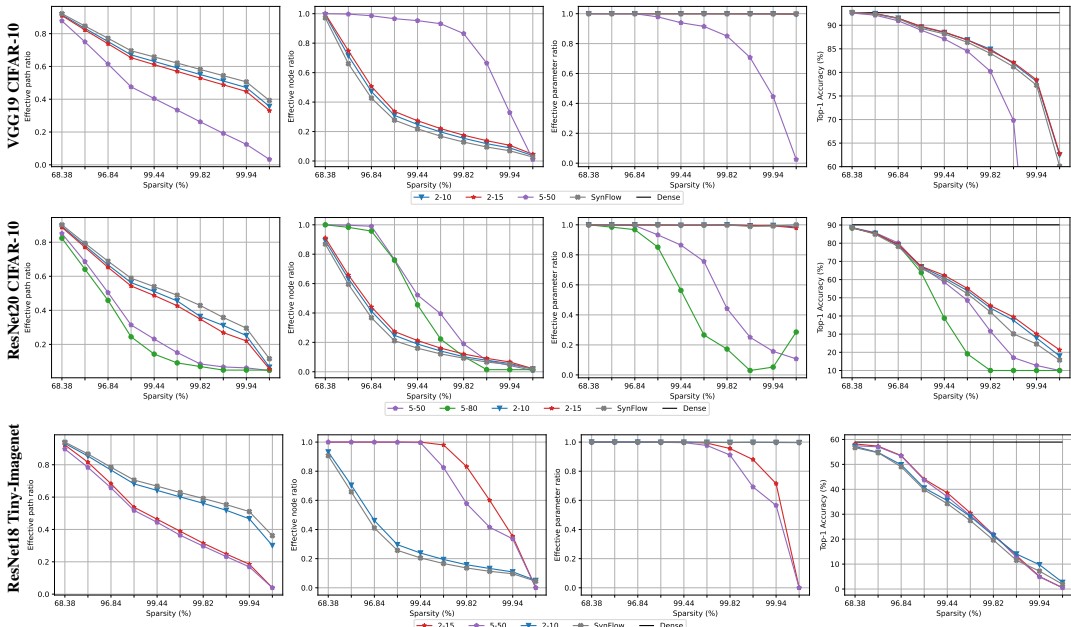

Figure 13: Scheduler ablation on different settings.

## H  POTENTIAL FUTURE DIRECTIONS

**Efficient Pruning Scheduler**    Based on the iterative pruning of SynFlow, we modify the pruning schedule with two additional hyperparameters, which balances better between effective paths and nodes. Adjusting these two drives to different architecture configurations as shown in Section 5.4. However, we believe that if we take into account a more sophisticated pruning scheduler, the pruned subnetwork will better satisfy the Node-Path Balancing Principle. From the view of the exploration and exploitation trade-off, it opens a new approach to pruning at initialization problems. To improve the effectiveness of the scheduler, we could borrow ideas from Reinforcement learning.

**Optimize the Node-Path Balancing**    From the Node-Path Balancing Principle, our goal is to optimize the number of activated nodes and paths without using data given the sparsity. One possible way is that switch the pruning problem to a multi-objective optimization problem. In particular, our problem becomes *given an architecture, maximizing both effective paths and effective nodes with the constraint of the number of parameters*. This is a non-trivial problem, but with the long-standing development of optimization problems, there are many promising approaches.

**Proxy for Evaluate Networks in Neuron Architecture Search Problem**    Neural Architecture Search (NAS) is known as the process of automating architecture engineering which is the next step in the automation of machine learning Elsken et al. (2019). One main problem of NAS is computation overhead in estimating the performance of candidates Zoph & Le (2017); Zoph et al. (2018). Recently, Abdelfattah et al. (2021) leverage pruning methods perform zero-cost proxies. We believe our effective nodes and paths could be promising scores to evaluate the architecture candidates in NAS.

# I   TOY EXPERIMENT

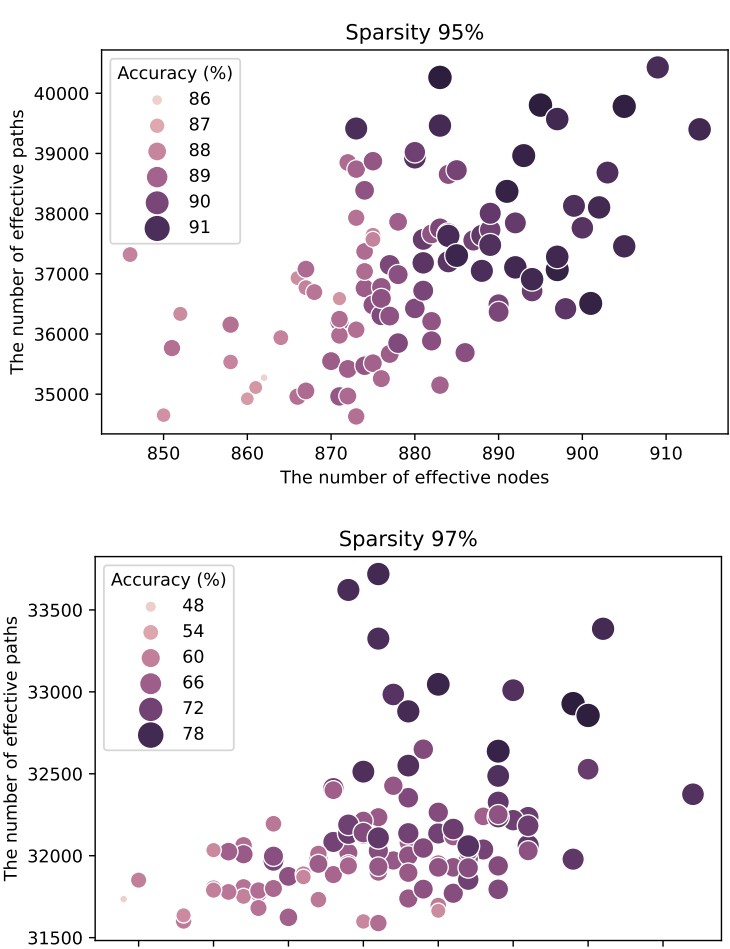

Figure 14: The toy experiments on a MLP network with three hidden layers and MNIST dataset. We sample 100 subnetworks with each two sparsity levels (95%, 97%)

In this experiment, we consider an MLP network with the numbers neurons in its layers are choosen as 784 - 128 - 64 - 10 with sparsity level of 95% and 97%, respectively. We keep all the nodes of input and output activated, other connections are randomly assigned into the network to create sparse subnetworks. Different subnetworks have different numbers of effective paths and nodes. We train all subnetworks with the same procedure to converge.

It is easy to see that subnetworks which have better balance between node and path tend to have better performance given the same sparsity. This proves that taking number of effective nodes and paths into account provides a novel insight in understanding PaI methods and then open new research directions in designing network architectures.

## J    COMPARISON WITH RAREGEM AND LTH

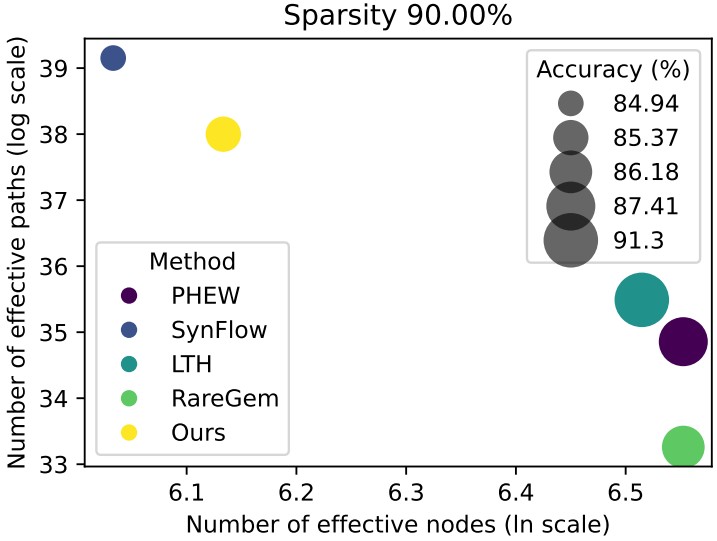

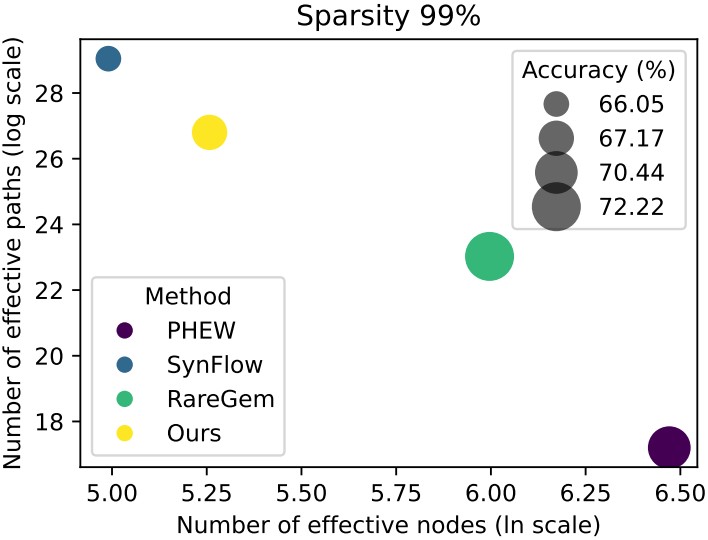

Figure 15: Comparison with RareGem and LTH methods on three aspects the number of effective nodes, paths and accuracy after training subnetworks.

We reproduce the results of LTH (Frankle & Carbin, 2018) and RareGem (Sreenivasan et al., 2022) on CIFAR-10 with Resnet20 to figure out whether subnetworks produced by these methods support our proposed principle or not. We compute the number of effective nodes and paths of subnetworks generated by PHEW, SynFlow, LTH, RareGem, and Ours. We then visualize in the Figure 15. We consider LTH as a upper bound and a optimal solution for balancing node-path. Through the additional experiments, we observe that subnetwork near optimal solution in terms of effective nodes and paths show better performance (accuracy after training). In extreme sparsity levels, RareGem produce better node-path balanced subnetworks compared to others, leading to higher performance of subnetworks. We strongly believe these results align with our principle and further support our principle as a necessary condition for good PaI. We want to highlight that RareGem and LTH both use dataset to produce subnetworks, and RareGem is 2x costly compared with data-agnostic PaI methods since it requires a "gem mining" process.

