# OpenReview forum: "Understanding Pruning at Initialization: An Effective Node-Path Balancing Perspective"
_ICLR.cc/2023/Conference — Submitted to ICLR 2023_

### Official Review · Reviewer_4Ufw · 2022-10-23

**Confidence:** 3
**Correctness:** 2
**Technical Novelty And Significance:** 2
**Empirical Novelty And Significance:** 2
**Recommendation:** 3

**Clarity, Quality, Novelty And Reproducibility:**

* The paper writing is okay, but needs some improvement.
* This paper is providing a novel approach, but the algorithm is not that new.





**Strength And Weaknesses:**

* Strength
    * This paper focuses on an interesting problem of analyzing the intriguing property of PaI methods.

* Weakness
    * The conjecture made by the authors is not supported by theory.
    * The empirical results are not supported by theory. For example, the empirical observation talks about two regimes (the normal sparsity and the extreme sparsity), but we do not know why 99% is a good threshold making different behaviors.
    * Algorithm 1 is proposed based on the conjecture, but I would say it is a small variant of Synflow and not optimized for the purpose of balancing two metrics. Does the success of Algo 1 imply that we really need the balance of two metrics? I see only little connection btw Node-path balancing principle and the design of algorithm 1.


* Question

> Page 5, "These observations indicate that increasing the number of effective paths (SynFlow) or effective nodes (PHEW) alone is not sufficient in the design of PaI methods."

I didn't fully get why the observation in Fig.2 implies this. From the figure, we cannot say SynFlow and PHEW are not a good option, I guess?

> Page 6, "From observations in Section 3.3 where both effective paths and nodes play critical roles in performances of subnetworks."

I guess this is not grammatically correct. Maybe the authors mean "From observations in Section 3.3, both effective paths and nodes play critical roles in performances of subnetworks"?

> Algorithm 1 line 6

Isn't it $t < T_{max}$, to be consistent with the description "in the first Tmax pruning iteration, we use random pruning after each (∆t − 1) steps"?

> Is node-path balancing principle necessary for having a good performance?

A recent PaI method [R1] passes the sanity check of layer wise shuffling and performs similar or better than IMP. I am curious whether this method has balanced node/path. If [R1] does not have balanced node/path, maybe the node-path balancing principle is not a necessary condition for good PaI?

[R1] Rare Gems: Finding Lottery Tickets at Initialization

**Summary Of The Paper:**

Motivated by a recent findings on the properties of prune at initialization (Su et al., 2020, Frankle et al., 2021), this paper proposes a new way of studying PaI methods. It suggests new metrics -- the number of effective paths" and "the number of effective nodes" -- which are proxies to the performance of pruned network according to the authors' claim. The authors conjecture that a good balance between these two metrics are needed to let the pruned network have a good performance. Motivated by this so-called "node-path balancing principle", this paper suggests a simple method to improve SynFlow.

**Summary Of The Review:**

This paper is suggesting new direction of understanding PaI in terms of effective node/path in the network, but we cannot have much insight or logic from their discussion & results.

---

> ### Author Response · Authors · 2022-11-17
> **Responding to Reviewer 4Ufw's comments part 1/2**
>
> We thank reviewer 4Ufw for valuable and constructive feedback. We provide our responses below.
>
> **Q1. The conjecture made by the authors is not supported by theory.**
>
> We agree that our paper is not a theoretical study. However, the underlying motivations for our method are evident and easy to understand. Let us recall that focusing solely on optimization of either nodes or paths will lead to a loss on the remaining quantity, which leads to suboptimal performance. This intuition is clear, and we think that unnecessary theoretical analysis would add little to our paper at this state. The theoretical problems of identifying the balancing points and the precise relation between node-path and performance are difficult research problems in their own rights, and we postpone them to future works.
>
>
> **Q2. The empirical results are not supported by theory. For example, the empirical observation talks about two regimes (the normal sparsity and the extreme sparsity), but we do not know why 99% is a good threshold making different behaviors.**
>
> We follow the setting of Cho et. al., 2021 and Price and Tanner et. al., 2021 to consider 99% as a threshold. When sparsity approaches 99%, subnetworks become more vulnerable with shuffling (as shown in section 3.3) or arranging connections (as observed Tanaka et. al., 2020) since it is easy to lead to layer collapse. However, it is true that this threshold may be different with different architecture, but to make it easy to follow, we assume 99% as a good threshold.
>
> **Q3. Algorithm 1 is proposed based on the conjecture, but I would say it is a small variant of Synflow and not optimized for the purpose of balancing two metrics. Does the success of Algo 1 imply that we really need the balance of two metrics? I see only little connection btw Node-path balancing principle and the design of algo 1.**
>
> We agree that the Algo. 1 is not optimized for the purpose of balancing two metrics. Through Algo. 1 we want to emphasize that even a slight modification of Synflow, that helps to balance the numbers of effective nodes and paths better, will lead to better performance of pruned subnetworks. This is especially clear in the extreme sparsity regime.
> In Algo 1, we introduce two additional hyperparameters which implicitly control the balance between node and path. We refer the reviewer to our answer to Q2 of reviewer uZiV and Section 5.3 for more details.
> We believe that Algo. 1 provides a good evidence that we should take into account both node and path in designing PaI methods.
>
> **Q4. Page 5, "These observations indicate that increasing the number of effective paths (SynFlow) or effective nodes (PHEW) alone is not sufficient in the design of PaI methods." I didn't fully get why the observation in Fig.2 implies it's statement. From the figure, we cannot say SynFlow and PHEW are not a good option, I guess?**
>
> We agree with the reviewer that we can not say SynFlow or PHEW are not a good option. Through Section 3.3, we would like to highlight that designing the sparse network at initialization phase should focus both on the number of effective paths and nodes. We empirically show that shuffling SynFlow or PHEW shows competitive performance despite of the changes in the configuration of subnetworks in the regular sparsities. In particular, shuffling SynFlow increase(s) the width of subnetworks while the effective paths decrease compared with the original version, however, the performance is still comparable or even higher. This indicates that maximizing effective paths methods like SynFlow are not efficient. Besides, when sparsity levels go to extreme cases, although PHEW produces wider subnetworks than SynFlow, spare networks produced by PHEW have lower performance than SynFlow due to the lack of effective input-output paths. These observations indicate that we should take into account both the numbers of effective nodes and paths when designing PaI methods.
>
>
>
> Refs:
>
> Price, Ilan, and Jared Tanner. "Dense for the price of sparse: Improved performance of sparsely initialized networks via a subspace offset." International Conference on Machine Learning. PMLR, 2021.
>
> Cho, Minsu, Ameya Joshi, and Chinmay Hegde. "Espn: Extremely sparse pruned networks." 2021 IEEE Data Science and Learning Workshop (DSLW). IEEE, 2021.
>
> Tanaka, Hidenori, et al. "Pruning neural networks without any data by iteratively conserving synaptic flow." Advances in Neural Information Processing Systems 33 (2020): 6377-6389.

---

> > ### Author Response · Authors · 2022-11-17
> > **Responding to Reviewer 4Ufw's comments part 2/2**
> >
> > **Q5. Page 6, "From observations in Section 3.3 where both effective paths and nodes play critical roles in performances of subnetworks." I guess this is not grammatically correct.**
> >
> > We thank the reviewer for addressing our error, we have modified it and highlighted it in blue in the updated manuscript.
> >
> >
> > **Q6. Isn’t it $t < T_\text{max}$ to be consistent with the description “in the first Tmax we use random pruning after each ($\Delta_t - 1$) steps”**
> >
> > In Algorithm 1 line 6 we set the conditions are $t \bmod \Delta_t = 0$ and $t < T_\text{max}$, and we believe these conditions align with our description.
> >
> >
> > **Q7. A recent PaI method (Rare Gem) passes the sanity check of layer wise shuffling and performs similar or better than IMP. I am curious whether this method has balanced node/path. If Rare Gem does not have balanced node/path, maybe the node-path balancing principle is not a necessary condition for good PaI?**
> >
> > We would like to thank the reviewer for the suggestion. We have run RareGem and Lottery Ticket Hypothesis (LTH) as well to measure metrics. We consider LTH as a upper bound and a optimal solution for balancing node-path. Through the additional experiments, we observe that subnetwork near optimal solution in terms of effective nodes and paths show better performance (accuracy after training), please refer to Appendix J for visualization in updated manuscript. In extreme sparsity levels, RareGem produce better node-path balanced subnetworks compared to others, leading to higher performance of subnetworks. We strongly believe these results align with our principle and further support our principle as a necessary condition for good PaI. We want to highlight that RareGem and LTH both use dataset to produce subnetworks, and RareGem is 2x costly compared with data-agnostic PaI methods since it requires a "gem mining” process. Due to the time limit we do not report the result of LTH in extreme cases, besides, in the Rare Gem paper the authors have shown that it is better than LTH in intensive sparsity levels. So we believe by only comparing with RareGem in extreme cases is sufficient to support our hypothesis.
> >
> > | Sparsity=90.00% | Node |   Path  |  Acc  |
> > |--------------|:----:|:-------:|:-----:|
> > | PHEW         |  701 | 7.18e34 | 87.41 |
> > | SynFlow      |  417 | 1.42e39 | 84.94 |
> > | LTH          |  675 | 3.09e35 | 91.30 |
> > | RareGem      |  701 | 1.82e33 | 86.18 |
> > | Ours         |  461 | 9.98e37 | 85.37 |
> >
> > | Sparsity=96.84% | Node |   Path  |  Acc  |
> > |-----------------|:----:|:-------:|:-----:|
> > | PHEW            |  700 | 4.98e25 | 81.94 |
> > | SynFlow         |  257 | 9.24e33 | 78.22 |
> > | LTH             |  655 | 1.12e26 | 86.23 |
> > | RareGem         |  616 | 7.21e27 | 83.36 |
> > | Ours            |  309 | 1.48e32 | 79.18 |
> >
> > | Sparsity=99.00% | Node |   Path  |   Acc  |
> > |--------------|:----:|:-------:|:------:|
> > | PHEW         |  646 | 1.59e17 |  70.44 |
> > | SynFlow      |  147 | 1.11e29 | 66.046 |
> > | LTH          |   _  |    _    |    _   |
> > | RareGem      |  402 | 1.06e23 |  72.22 |
> > | Ours         |  192 | 6.33e26 |  67.17 |
> >
> > | Sparsity=99.44% | Node |   Path  |  Acc  |
> > |-----------------|:----:|:-------:|:-----:|
> > | PHEW            |  554 | 4.40e13 | 63.33 |
> > | SynFlow         |  112 | 3.98e26 |   60.06  |
> > | LTH             |   _  |    _    |   _   |
> > | RareGem         |  363 | 1.10e16 | 66.65 |
> > | Ours            |  149 | 1.10e24 | 62.28 |
> >
> > Ref:
> > Sreenivasan, Kartik, et al. "Rare Gems: Finding Lottery Tickets at Initialization." Advances in Neural Information Processing Systems 35 (2022).

---

> > ### Comment · Reviewer_4Ufw · 2022-11-21
> > **Reply to the author response**
> >
> > Thanks for the response as well as additional experiments. Although some of my concerns are handled, I still believe this empirical observation needs more theoretical justification. I will maintain my score.

---

### Official Review · Reviewer_8VAP · 2022-10-24

**Confidence:** 3
**Correctness:** 2
**Technical Novelty And Significance:** 3
**Empirical Novelty And Significance:** 2
**Recommendation:** 5

**Clarity, Quality, Novelty And Reproducibility:**

The key idea of this paper to think of pruning at initialization as a two objective problem is sufficiently novel and somewhat justified in the context of prior work. The main flaw of this paper is that it does not do enough to convince the reader that these two objectives can indeed be used to understand pruning at initialization.

**Strength And Weaknesses:**

Strengths:
- The idea of dividing the pruning at initialization problem into these two objectives is novel
- The paper is mostly easy to follow

Weaknesses:
- The new objectives are not supported by much more than heuristic arguments
- The explanation in section 3.3 that attempts to explain the layerwise shuffling phenomenon through the lens of these two objectives is underdeveloped and should be expanded to allow readers to understand how the trade-off between the two objectives explains these results
- The experimental results where the paper attempts a first pass at using these two objectives to prune at initialization are unconvincing (understandably the authors do clarify that this is not the goal of the paper, but considering lack of theoretical justification for the objectives, the experiments should instead instill faith in these two objectives)


**Summary Of The Paper:**

This paper investigates pruning at initialization through the lens of 'effective nodes' and 'effective paths'. Effective paths are defined as paths that exist from input node to output node and effective nodes as nodes that participate in at least one of these paths. The paper argues that these 2 objectives must be simultaneously met if one is to successfully prune at initialization.

**Summary Of The Review:**

The paper presents a novel idea for pruning at initialization in the form of the two objectives, however, lack of theoretical justification or strong empirical results makes the proposal unconvincing. Perhaps the authors could supplement the experiments with simpler toy experiments where they can explicitly optimize for the two objectives, this might help convince readers of the usefulness of the objectives.

---

> ### Author Response · Authors · 2022-11-17
> **Responding to Reviewer 8VAP's comments**
>
> We thank reviewer 8VAP for valuable and constructive feedback. We provide our responses below.
>
> **Q1. The new objectives are not supported by much more than heuristic arguments**
>
> Our paper is purely experimental. Our main goal is to demonstrate that there is a need to rethink the concept we currently rely on to design and explain the efficiency of sparse neural networks, which typically focuses on increasing the number of effective nodes. We show via a number of experiments that there is a need to consider the number of effective nodes as well. In addition, by just making a slight change to SynFlow in a very naive way to force it considering both effective nodes and paths,  we demonstrate that the performance significantly increases. Thus, we hope that our work can be used as a justification for further theoretical analysis and rethinking of the way we do PaI.
>
> **Q2. The explanation in section 3.3 that attempts to explain the layerwise shuffling phenomenon through the lens of these two objectives is underdeveloped and should be expanded to allow readers to understand how the trade-off between the two objectives explains these results**
>
> Firstly, we try to confirm the finding of Frankle et al. 2021 that shuffling does not decrease the performance of subnetworks. Through the lens of effective nodes and paths, we try to explain why these happen. In particular, when shuffling the number of effective nodes increases significantly in cases of SNIP and SynFlow (methods focus on maximizing connectivity). These shuffling versions of subnetworks generated by SNIP and SynFlow become similar to PHEW's subnetwork in terms of the number of effective nodes and paths.
>
> | Sparsity=68.38%    | Node |   Path  |  Acc  |
> |-----------------|:----:|:-------:|:-----:|
> | SNIP            |  611 | 6.76e43 | 87.88 |
> | Shuffle SNIP    |  701 | 2.74e41 | 88.58 |
> | SynFlow         |  609 | 3.26e44 | 88.64 |
> | Shuffle SynFlow |  701 | 4.47e42 | 88.73 |
> | PHEW            |  701 | 1.78e43 | 90.47 |
> | Shuffle PHEW    |  701 | 1.70e43 | 90.31 |
>
> | Sparsity=90.00%    | Node |   Path  |  Acc  |
> |-----------------|:----:|:-------:|:-----:|
> | SNIP            |  473 | 4.25e37 | 84.02 |
> | Shuffle SNIP    |  701 | 1.61e33 | 85.61 |
> | SynFlow         |  417 | 1.42e39 | 84.94 |
> | Shuffle SynFlow |  701 | 1.98e34 | 86.08 |
> | PHEW            |  701 | 7.20e34 | 87.45 |
> | Shuffle PHEW    |  701 | 5.33e34 | 87.31 |
>
> Secondly, the findings of Frankle et al. 2021 are inconsistent for the extreme sparsity regime where sparsity levels are higher than 99%, due to the significant decrease of both numbers of effective paths and nodes in subnetworks after shuffling.
>
> | Sparsity=99.68% | Node |   Path  |  Acc  |
> |-----------------|:----:|:-------:|:-----:|
> | SNIP            |  117 | 1.44e12 | 38.41 |
> | Shuffle SNIP    |  216 |  5.69e5 | 44.23 |
> | SynFlow         | 85 | 1.28e24 | 52.26 |
> | Shuffle SynFlow |  278 |  6.91e6 | 41.13 |
> | PHEW            |  431 | 4.21e10 | 53.48 |
> | Shuffle PHEW    |  225 |  5.50e7 | 45.35 |
>
>
> **Q3. The experimental results where the paper attempts a first pass at using these two objectives to prune at initialization are unconvincing (understandably the authors do clarify that this is not the goal of the paper, but considering lack of theoretical justification for the objectives, the experiments should instead instill faith in these two objectives)**
>
> We agree that the paper’s main goal is not to provide theoretical analysis of the idea, but instead, we experimentally provide evidence that there is a need to consider both effective nodes and paths.
>
> **Q4. The main flaw of this paper is that it does not do enough to convince the reader that these two objectives can indeed be used to understand pruning at initialization. Perhaps the authors could supplement the experiments with simpler toy experiments where they can explicitly optimize for the two objectives, this might help convince readers of the usefulness of the objectives.**
>
> We have run a toy experiment as follows: Consider a MLP network 784 - 128 - 64 - 10 with sparsity level of 95% and 97%, respectively. We keep all the nodes of input and output activated, other connections are randomly assigned into the network to create subnetworks. Different subnetworks will have different numbers of effective paths and nodes. We train all subnetworks with the same procedure to convergence. Please refer to Figure 1 in the Introduction section and Appendix I in the updated version for more details of the toy experiment. In Figure 14 in Appendix I, we observe that subnetworks which have a better balance between effective nodes and paths tend to have better performance given the same sparsity. This indicates that taking number of effective nodes and paths into account provides a novel insight in understanding PaI methods and opens new research directions in designing network architectures.
>
> Ref:
> Frankle et. al., Pruning neural networks at initialization: Why are we missing the mark? ICLR 2021.

---

> > ### Comment · Reviewer_8VAP · 2022-11-21
> > **Experiments not convincing enough for an empirical paper**
> >
> > For an empirical paper, the experiment section is not convincing enough. The paper does propose interesting and novel ideas, but without a way to evaluate these ideas theoretically or empirically, it is difficult to judge how useful these ideas actually would be to the field.
> >
> > I maintain my original assessment of this paper.
> >
> >  Recommendation: 5: marginally below the acceptance threshold

---

### Official Review · Reviewer_EqZe · 2022-10-31

**Confidence:** 4
**Correctness:** 3
**Technical Novelty And Significance:** 1
**Empirical Novelty And Significance:** 2
**Recommendation:** 3

**Clarity, Quality, Novelty And Reproducibility:**

The paper  is presented in a clear manner and thus delivers its point well. This work builds a lot on previous works and settings derived from them, so the ideas or experiments are not entirely novel or original.


**Strength And Weaknesses:**

Strength
- The writing is very clear, and the main idea and intuition behind are delivered well to readers.
- The intuition behind the proposed method makes sense and based upon previous research findings in the literature.
- The experiments are conducted systematically.

Weaknesses

- The proposed method introduces additional hyperparameters (\Delta_t and T_\text{max}) which can be suboptimally set in practice or expensive to perform. It is also hard to compare with others without solid fair hyperparameter searching. In terms of algorithmic or methodological advances it is simply considered as another schedule for iterative magnitude based pruning but with some resetting. When it comes to comparisons, it is hard to tell which one is better than which and things are not conclusive.
- The proposed method being not better than PHEW in terms of the subnet accuracy (for regular sparsity levels) weakens the main argument on the importance of node-path balancing, which is somewhat critical.
- Perhaps not the most critical issue, but it would have been better to add experiments on more diverse and larger data sets and network models since the paper relies on empirical evidence.
- The results on effective paths and effective nodes divided by regular and extreme sparsity levels seem somewhat inconsistent and remain based on hypothetical intuition level without direct or provable evidence. The divider threshold between can be thought of as favoring the interpretation of the proposed method without a clear setup as to how the threshold has to be chosen.


**Summary Of The Paper:**

The paper presents a new pruning approach called, node-path balancing principle, based on the intuition that both effective nodes and effective paths need to be preserved high such that training sparse neural networks from scratch can be performed well.
The method is essentially done by tuning the schedule of (random) pruning for the obtained connectivity in the subnetwork.
This intuition is obtained based on recent works on pruning including the shuffling mask effect.
The authors summarize their findings in two different regimes, regular and extreme sparsity levels, that while effective nodes are more important than effective paths for regular sparsity, it may not be the case for extreme sparsity levels.


**Summary Of The Review:**

I believe the paper investigates an important aspect based on good intuition. However, the methodology developed seems to be quite obvious, and the resulting algorithm is not backed up with strong empirical evidence.

---

> ### Author Response · Authors · 2022-11-17
> **Responding to Reviewer EqZe's comments**
>
> We thank reviewer EqZe for valuable and constructive feedback. We provide our responses below.
>
> **Q1. The proposed method introduces additional hyperparameters (\Delta_t and T_\text{max}) which can be suboptimally set in practice or expensive to perform. It is also hard to compare with others without solid fair hyperparameter searching. In terms of algorithmic or methodological advances it is simply considered as another schedule for iterative magnitude based pruning but with some resetting. When it comes to comparisons, it is hard to tell which one is better than which and things are not conclusive.**
>
> We agree that the proposed method based on two additional hyperparameters can be suboptimally set but the ultimate goal of this method is to underpin the proposed principle about node-path balancing.
>
> Regarding fair hyperparameter searching with others, instead of training the subnetworks with different $\Delta_t$ and $T_\text{max}$ to select the best subnetworks, we leverage our principle to select the subnetworks before training based on the number of effective nodes and paths. Since we do not have to train subnetworks, we believe the cost of hyperparameter searching is neglectable.
>
> We admit that comparing which one is better than the others is not straightforward. However, in this work, we would like to highlight our principle which opens a new research direction in pruning neural networks. More elaborated pruning methods based on this principle are worth pursuing research direction, but we postpone it to future work.
>
> **Q2. The proposed method being not better than PHEW in terms of the subnet accuracy (for regular sparsity levels) weakens the main argument on the importance of node-path balancing, which is somewhat critical.**
>
> We agree that our method is not better than PHEW in subnet accuracy in regular sparsity levels. However, we disagree with the reviewer that this weakens our argument on the importance of node-path balancing. In cases where PHEW outperforms our method, these subnetworks produced by PHEW are better at balancing between the number of effective nodes and paths. Our experimental results support the node-path balancing principle in designing PaI methods. Viewing PaI tasks as an optimization problem of balancing the effective nodes and paths given a sparsity level as a constraint, we can apply our principle as a guideline for the optimization process, in particular, maximizing both the number of effective nodes and paths.
>
> **Q3. Perhaps not the most critical issue, but it would have been better to add experiments on more diverse and larger data sets and network models since the paper relies on empirical evidence.**
>
> We agree with the reviewer that more diverse experimental settings will improve our paper, but due to the limited time, we postpone it to the next version of our paper.
>
>
> **Q4. The results on effective paths and effective nodes divided by regular and extreme sparsity levels seem somewhat inconsistent and remain based on hypothetical intuition level without direct or provable evidence. The divider threshold between can be thought of as favoring the interpretation of the proposed method without a clear setup as to how the threshold has to be chosen.**
>
> We follow the setting of Cho et. al., 2021 and Price et. al., 2021 to consider 99% as a threshold. When sparsity approaches 99%, subnetworks become more vulnerable to shuffling (as shown in Section 3.3) or arranging connections (as observed Tanaka et. al., 2020) since it is easy to lead to layer collapse. However, it is true that this threshold may be different with different architectures, but to make it easy to follow, we assume 99% as a good threshold.
>
> Refs:
>
> Price, Ilan, and Jared Tanner. "Dense for the price of sparse: Improved performance of sparsely initialized networks via a subspace offset." International Conference on Machine Learning. PMLR, 2021.
>
> Cho, Minsu, Ameya Joshi, and Chinmay Hegde. "Espn: Extremely sparse pruned networks." 2021 IEEE Data Science and Learning Workshop (DSLW). IEEE, 2021.
>
> Tanaka, Hidenori, et al. "Pruning neural networks without any data by iteratively conserving synaptic flow." Advances in Neural Information Processing Systems 33 (2020): 6377-6389.

---

### Official Review · Reviewer_uZiV · 2022-11-04

**Confidence:** 2
**Correctness:** 3
**Technical Novelty And Significance:** 2
**Empirical Novelty And Significance:** 2
**Recommendation:** 5

**Clarity, Quality, Novelty And Reproducibility:**

Clarity: Underlying motivation of the paper is clear and approaches suggested in the paper makes sense.
Quality/Novelty: Innovation in the paper sounds more empirical and less theoretical. Iterative pruning algorithm discussed in the paper doesn’t sound innovative. Adding more details on hyperparameters would have helped understand a few sections. However, findings in the paper would be very helpful to the researchers working on designing PaI methods.

**Strength And Weaknesses:**

Authors have well drafted the related work/background section, clearly outlining past work in neural network pruning, importance of nodes and connection configurations, extreme sparse network design. Empirical evaluation in the paper is very detailed - for example, sections on layerwise shuffling, impact on regular/extreme sparsities are intriguing.  Iterative pruning algorithm/Node-Path balancing principle described in the paper found to be very effective in optimizing the number of activated nodes and paths.

Iterative pruning algorithm to find a pruning mask described in the paper sounds to be similar to existing work in this literature and paper doesn’t sound innovative from a theoretical point. Some subsections seek more details - for example, what is the motivation to introduce additional hyperparameters described in section 4, and statements like “findings indicate that pruning neural networks in the regular sparsity regime should give more consideration to the number of effective neurons since the information flow is conserved by the sufficient number of input-output paths.”. Providing qualitative analysis around scheduler ablation experiments can help understand the empirical numbers better.

**Summary Of The Paper:**

Neural network pruning (also referred as pruning at initialization(PaI)) task involves balancing the tradeoff between model complexity and accuracy. This makes it a challenging task. In this paper, authors have proposed a few guidelines to design PaI methods that involve leveraging configuration of subnetwork 1) number of effective paths 2) number of effective nodes. Authors provide new insights to the working mechanism of PaI methods, and open new research directions on neural network pruning methods. Authors find that width of subnetworks and node-path balancing plays an important role in designing PaI subnetworks. Experiments were conducted using PaI methods like SNIP, SynFlow, and PHEW on different architectures( ResNet, VGG) and datasets (CIFAR, Tiny-Imagenet).

**Summary Of The Review:**

This paper provides helpful empirical insights for designing pruning at initialization(PaI) methods. Experiments show the impact of the topology of subnetworks - specifically subnetwork configurations like 1) number of effective paths 2) number of effective nodes in designing PaI methods, and how to balance these metrics in regular sparsity vs extremely sparse scenarios. Experiments were conducted using PaI methods like SNIP, SynFlow, and PHEW on different architectures( ResNet, VGG) and datasets (CIFAR, Tiny-Imagenet). Please refer to strengths/weaknesses section for more details.

---

> ### Author Response · Authors · 2022-11-17
> **Responding to Reviewer uZiV's comments part 1/2**
>
> We thank reviewer uZiV for support and constructive feedback to improve our work. We address concerns and questions of the reviewer below.
>
> **Q1. Iterative pruning algorithm to find a pruning mask described in the paper sounds to be similar to existing work in this literature**
>
> Indeed our pruning algorithm is a simple modification of Synflow, particularly, SynFlow focuses solely on optimizing the number of effective paths and neglects the number of effective nodes. We make simple modifications to increase the number of effective nodes and show that such modifications help balancing the numbers of effective nodes and paths, and improve the performance of pruned subnetworks compared to Synflow. This is also our main purpose of introducing this algorithm: even a slight modification of Synflow, which focuses on balancing the numbers of effective nodes and paths, results in a better performance of pruned networks. This phenomenon is demonstrated clearly in our experiments, especially in the extreme sparsity regime. The main message of our paper is that it provides empirical evidence that suggests using node path balancing as a principle to design new pruning algorithms. We will clarify this in our paper. Note that we do not position this paper as a comprehensive theoretical work, but rather an empirical one which reveals a very important new phenomenon.
>
> **Q2. What is the motivation to introduce additional hyperparameters and providing qualitative analysis around scheduler ablation experiments can help understand the empirical numbers better.**
>
> The extra parameters were introduced to control the trade-off between the numbers of effective nodes and paths. SynFlow only focuses on maximizing the number of effective paths which implicitly leads to narrow subnetworks. From the observation on Section 3.3 where shuffling connections in subnetworks obtained by SynFlow produces a trade-off between nodes and paths, and has a comparable performance in normal sparsity levels but fails in extreme sparsity cases. Therefore, we provide additional hyperparameters to better control the trade-off without leading to layer collapse in intensive sparsities. Intuitively, while randomly pruning the network when it is in normal sparsities may increase the subnetwork's width because it fortuitously prunes edges connected to high-degree nodes, a large $T_{max}$ easily drives subnetworks to layer-collapse since the pruner randomly removes important connections when the network is extremely sparse, which destroys a large number of input-to-output paths.
>
> We provide empirical results on three sparsities on the setting ResNet20-CIFAR10, through these results, we observe that:
> (i) Applying random pruning during the pruning process creates large-width subnetworks in regular sparsity levels.
> (ii) In extreme sparsity regime, a large $T_{max}$ significantly reduces the number of effective paths.
> (iii) Using random pruning more frequently in the early stage of pruning generates subnetworks with better performance in extreme sparsities.
>
> *Sparsity 90.00%*
> | $T_{max}$ | $\Delta_t$ | Node |   Path  | Accuracy |
> |-----------|------------|:----:|:-------:|:--------:|
> | 10        | 2          |  444 | 3.07e38 |   85.41  |
> | 15        | 2          |  461 | 9.87e37 |   85.39  |
> | 30        | 5          |  701 | 4.81e34 |   86.39  |
> | 50        | 5          |  699 | 6.99e33 |   85.82  |
> | 80        | 5          |  689 | 4.21e31 |   85.46  |
> | SynFlow   |            |  417 | 1.42e39 |   84.94  |
>
> *Sparsity 99.00%*
> | $T_{max}$ | $\Delta_t$ | Node |   Path  | Accuracy |
> |-----------|------------|:----:|:-------:|:--------:|
> | 10        | 2          |  174 | 5.80e27 |   67.02  |
> | 15        | 2          |  192 | 6.29e26 |   67.25  |
> | 30        | 5          |  588 | 5.86e15 |   67.53  |
> | 50        | 5          |  535 | 3.18e15 |   66.96  |
> | 80        | 5          |  532 | 1.14e12 |   63.82  |
> | SynFlow   |            |  148 | 1.11e29 |   66.05  |
>
> *Sparsity 99.68%*
> | $T_{max}$ | $\Delta_t$ | Node |   Path  | Accuracy |
> |-----------|------------|:----:|:-------:|:--------:|
> | 10        | 2          |  99  | 2.91e22 |   53.92  |
> | 15        | 2          |  112 | 1.15e21 |   55.13  |
> | 30        | 5          |  220 |  3.30e7 |   45.33  |
> | 50        | 5          |  277 |  3.22e7 |   48.55  |
> | 80        | 5          |  156 |  32032  |   19.14  |
> | SynFlow   |            | 85.2 | 1.28e24 |   52.26  |

---

> > ### Author Response · Authors · 2022-11-17
> > **Responding to Reviewer uZiV's comments part 2/2**
> >
> > **Q3. Details on the statement: "findings indicate that pruning neural networks in the regular sparsity regime should give more consideration to the number of effective neurons since the information flow is conserved by the sufficient number of input-output paths"**
> >
> > As mentioned in Section 2, Price et. al., 2021 used a fixed dense network and a set of trainable weights to learn the task. Their goal is to preserve the information flow from input to output, reducing the number of parameters to perform tasks effectively. Therefore, in PaI, when subnetworks are at regular sparsity levels, the number of effective paths is still large enough to preserve the information flow. This implies that in these cases, we should pay more attention to optimize the number of effective nodes to achieve the node-path balance.
> >
> > Ref: Price, Ilan, and Jared Tanner. "Dense for the price of sparse: Improved performance of sparsely initialized networks via a subspace offset." International Conference on Machine Learning. PMLR, 2021.
> >
> > **Q4. Paper doesn’t sound innovative from a theoretical point**
> >
> > We agree that our paper is not a theoretical study. However, the underlying motivations for our method are justified and easy to understand. Let us recall that focusing on optimization of either nodes or paths will lead to a loss on the remaining quantity, which leads to suboptimal performance. This statement is intuitive and we think that unnecessary theoretical analysis would add little to our paper at this stage. The key theoretical challenge here is the problem of identifying the balancing points and the precise relation between node-path and performance. These are difficult research problems in their own rights, and we postpone them to future work.

---

### Decision · Program_Chairs · 2023-01-20

**Decision:**

Reject

**Justification For Why Not Higher Score:**

Explained in part 1.

**Justification For Why Not Lower Score:**

N/A

**Metareview: Summary, Strengths And Weaknesses:**

This paper investigates pruning at initialization, and demonstrate that effective paths (paths that exist from input node to output node) and effective nodes (nodes that participate in at least one of these paths, must be simultaneously met to successfully prune a network at initialization. The paper has some novel ideas which have not been explored before, and fits well in the context of prior work. However, the main concern is that the paper cannot well justify that preserving the proposed metrics ensure the performance of the pruned network. The paper will be strengthen by either supporting the proposed technique with theoretical justification or adding more extensive experiments to provide deeper understanding of the effect of the proposed metrics on the performance of network pruned at initialization.